# Subtype-specific responses of hKv7.4 and hKv7.5 channels to polyunsaturated fatty acids reveal an unconventional modulatory site and mechanism

**Damon JA Frampton[1], Koushik Choudhury[2], Johan Nikesjö[1], Lucie Delemotte[2], Sara I Liin[1]\***

[1]Department of Biomedical and Clinical Sciences, Linköping University, Linköping, Sweden; [2]Science for Life Laboratory, Department of Applied Physics, KTH Royal Institute of Technology, Solna, Sweden

**Abstract** The $K_V7.4$ and $K_V7.5$ subtypes of voltage-gated potassium channels play a role in important physiological processes such as sound amplification in the cochlea and adjusting vascular smooth muscle tone. Therefore, the mechanisms that regulate $K_V7.4$ and $K_V7.5$ channel function are of interest. Here, we study the effect of polyunsaturated fatty acids (PUFAs) on human $K_V7.4$ and $K_V7.5$ channels expressed in *Xenopus* oocytes. We report that PUFAs facilitate activation of $hK_V7.5$ by shifting the $V_{50}$ of the conductance *versus* voltage (G(V)) curve toward more negative voltages. This response depends on the head group charge, as an uncharged PUFA analogue has no effect and a positively charged PUFA analogue induces positive $V_{50}$ shifts. In contrast, PUFAs inhibit activation of $hK_V7.4$ by shifting $V_{50}$ toward more positive voltages. No effect on $V_{50}$ of $hK_V7.4$ is observed by an uncharged or a positively charged PUFA analogue. Thus, the $hK_V7.5$ channel's response to PUFAs is analogous to the one previously observed in $hK_V7.1$–7.3 channels, whereas the $hK_V7.4$ channel response is opposite, revealing subtype-specific responses to PUFAs. We identify a unique inner PUFA interaction site in the voltage-sensing domain of $hK_V7.4$ underlying the PUFA response, revealing an unconventional mechanism of modulation of $hK_V7.4$ by PUFAs.

**\*For correspondence:** sara.liin@liu.se

## Editor's evaluation

In this manuscript, the authors describe the effects of polyunsaturated fatty acids in voltage-gated potassium channels of the Kv7 family that are specific for each subtype. The authors uncover a mechanism for this specificity which suggests that subtle structural differences can account for large effects, contributing to the physiological functions of these ion channels.

## Introduction

$K_V7$ voltage-gated potassium channels are expressed in several tissues, where they serve to attenuate excitability by conducting an outward $K^+$ current. The five members of the family, $K_V7.1$-$K_V7.5$ (encoded by *KCNQ1-KCNQ5* genes), are important for human physiology, which is often emphasized in diseased states caused by dysfunctional channels. For instance, $K_V7.1$ is predominantly expressed in cardiomyocytes, and mutations to this channel are a known risk factor for developing cardiac arrhythmia (*Barhanin et al., 1996*; *Brewer et al., 2020*; *Sanguinetti et al., 1996*; *Tester and Ackerman, 2014*). $K_V7.2$ and $K_V7.3$ are broadly expressed in neurons, where they form heterotetrameric $K_V7.2/7.3$ channels, and mutations to these channels may give rise to epilepsy or chronic pain (*Biervert et al., 1998*;

**eLife digest** In order to carry out their roles in the body, cells need to send and receive electrical signals. They can do this by allowing ions to move in and out through dedicated pore-like structures studded through their membrane. These channels are specific to one type of ions, and their activity – whether they open or close – is carefully controlled. In humans, defective ion channels are associated with conditions such as irregular heartbeats, epileptic seizures or hearing loss.

Research has identified molecules known as polyunsaturated fatty acids as being able to control the activity of certain members of the $K_V7$ family of potassium ion channels. The $K_V7.1$ and $K_V7.2/7.3$ channels are respectively present in the heart and the brain; $K_V7.4$ is important for hearing, while $K_V7.5$ plays a key role in regulating muscle tone in blood vessels. Polyunsaturated fatty acids can activate $K_V7.1$ and $K_V7.2/7.3$ but their impact on $K_V7.4$ and $K_V7.5$ remains unclear.

Frampton et al. explored this question by studying human $K_V7.4$ and $K_V7.5$ channels expressed in frog egg cells. This showed that fatty acids activated $K_V7.5$ (as for $K_V7.1$ and $K_V7.2/7.3$), but that they reduced the activity of $K_V7.4$.

Closely examining the structure of $K_V7.4$ revealed that the fatty acids were binding to a different region compared to the other $K_V7$ channels. When this site was made inaccessible, fatty acids increased the activity of $K_V7.4$, just as for the rest of the family.

These results may help to understand the role of polyunsaturated fatty acids in the body. In addition, knowing how these molecules interact with channels in the same family will be useful for optimising a drug's structure to avoid side effects. However, further research will be needed to understand the broader impact in a more complex biological organism.

---

*Nappi et al., 2020*; *Wang et al., 1998*). $K_V7.4$ is of particular importance in the auditory system where it forms homotetrameric channels responsible for a $K^+$ conductance at the resting membrane potential of cochlear outer hair cells (OHCs) (*Kharkovets et al., 2000*; *Kubisch et al., 1999*; *Rim et al., 2021*). Mutations that perturb the trafficking or function of $K_V7.4$ in OHCs are associated with a subtype of progressive hearing loss known as DFNA2 (*Gao et al., 2013*; *Kharkovets et al., 2006*; *Kubisch et al., 1999*; *Rim et al., 2021*). $K_V7.5$ is more widely spread through the central nervous system and is particularly important in regulating excitability in the hippocampus (*Fidzinski et al., 2015*; *Schroeder et al., 2000*; *Tzingounis et al., 2010*). $K_V7.1$, $K_V7.4$, and $K_V7.5$ are also expressed in various smooth muscle cells, such as vascular smooth muscle cells (VSMC), suggesting that several $K_V7$ subtypes, including heterotetrameric $K_V7.4/7.5$ channels, may contribute to the hyperpolarizing $K^+$ current in VSMCs that promotes vasodilation by preventing $Ca^{2+}$-dependent contraction (*Mani et al., 2013*; *Ng et al., 2011*; *Stott et al., 2014*; *Bercea et al., 2021*). The modulation of $K_V7.1$ and $K_V7.2/7.3$ channels by endogenous and pharmacological compounds has been extensively studied (*Miceli et al., 2018*; *Wu and Larsson, 2020*). However, less is known about the modulation of $K_V7.4$ and $K_V7.5$. Here, we studied the effects of a class of channel modulators, polyunsaturated fatty acids (PUFAs), on $K_V7.4$ and $K_V7.5$.

There is emerging evidence that suggests that PUFAs influence the physiology of tissues that express $K_V7.4$ and $K_V7.5$ channels. For instance, a number of studies have found an inverse relationship between hearing loss and the PUFA plasma concentration, suggesting that the risk of impaired hearing decreases with an increased dietary intake of $\omega-3$ PUFAs such as docosahexaenoic acid (DHA) and eicosapentaenoic acid (EPA) (*Curhan et al., 2014*; *Dullemeijer et al., 2010*; *Gopinath et al., 2010*). Meta-analyses of randomized control trials have found that a multitude of beneficial cardiovascular outcomes, including antiinflammatory and hypotensive effects, are associated with an increased intake of PUFAs (*AbuMweis et al., 2018*; *Miller et al., 2014*). There are likely several mechanisms that contribute to these PUFA effects. For instance, the protective PUFA effect on hearing loss has been attributed to cerebrovascular effects, reasoning that PUFAs improve circulation to the cochlea (*Curhan et al., 2014*; *Dullemeijer et al., 2010*; *Gopinath et al., 2010*). PUFA-induced vasodilation has in part been attributed to the activation of $Ca^{2+}$-dependent and ATP sensitive $K^+$ channels by PUFAs (*Hoshi et al., 2013*; *Limbu et al., 2018*; *Bercea et al., 2021*). However, little is known about the putative direct contribution of $K_V7.4$ and $K_V7.5$ channels to PUFA effects. We find this open question interesting because PUFAs have been shown to facilitate activation of both $K_V7.1$ and $K_V7.2/7.3$ channels (*Bohannon et al., 2019*; *Larsson et al., 2020*; *Liin et al., 2015*; *Liin et al., 2016*; *Taylor and*

*Sanders, 2017*). This PUFA-induced facilitation of activation is mediated through a lipoelectric mechanism in which the PUFA tail inserts into the outer leaflet of the lipid bilayer adjacent to the channel, whereupon the negatively charged carboxyl head group of the PUFA interacts electrostatically with positively charged arginines in the upper half of the voltage-sensing domain (VSD) of the channel (*Liin et al., 2018*; *Yazdi et al., 2021*). This electrostatic interaction facilitates the outward movement of the S4 helix, causing a shifted voltage dependence of channel opening toward more negative voltages. However, the effect of PUFAs on $K_V7.4$ and $K_V7.5$ remains unstudied. In this study, we therefore aimed to characterize the response of human $K_V7.4$ and $K_V7.5$ channels (henceforth referred to as $hK_V7.5$ or $hK_V7.4$) to PUFAs, in order to expand our understanding of how the $K_V7$ family of channels responds to these lipids.

We report that PUFAs facilitate activation of $hK_V7.5$ by shifting the voltage dependence of channel opening toward more negative voltages. Surprisingly, we find that PUFAs inhibit activation of the $hK_V7.4$ channel by shifting the voltage dependence of channel opening toward more positive voltages. Thus, the $hK_V7.5$ channel's response to PUFAs is largely in line with the responses that have previously been observed in $hK_V7.1$ and $hK_V7.2/7.3$, whereas the $hK_V7.4$ channel response is not. Providing a mechanistic explanation for this observation, we identify an unconventional inner PUFA site in the VSD of $hK_V7.4$ underlying PUFA-induced inhibition of the activation of $hK_V7.4$. Our study expands our understanding of how members of the $hK_V7$ family respond to PUFAs and reveal subtype specific responses and sites to these lipids.

## Results

### The PUFA docosahexaenoic acid facilitates activation of $hK_V7.5$, but inhibits activation of $hK_V7.4$

We began with investigating the effects of the physiologically abundant (*Kim et al., 2014*) PUFA DHA (molecular structure shown in *Figure 1A*) on homotetrameric $hK_V7.5$ or $hK_V7.4$ channels expressed in *Xenopus* oocytes. Activation of the $hK_V7.5$ channel was facilitated by 70 μM DHA, as was evident by the significant shift in the midpoint of the voltage dependence of channel opening ($V_{50}$) toward more negative voltages (*Figure 1B*, average shift of –21.5 ± 1.9 mV, p = <0.0001). This allows $hK_V7.5$ channels to open and conduct a $K^+$ current at more negative voltages in the presence of DHA. 70 μM DHA did not cause a consistent change in the maximum conductance ($G_{max}$) of the $hK_V7.5$ channel (average relative $\Delta G_{max}$ was 1.06 ± 0.11, p = 0.59).

In clear contrast to the facilitated activation observed for $hK_V7.5$, the activation of the $hK_V7.4$ channel was inhibited by 70 μM DHA, as was seen by the significant shift in the $V_{50}$ towards more positive voltages (*Figure 1C*, average shift of +11.8 ± 1.4 mV, p = <0.0001). Furthermore, the application of DHA led to a more shallow slope of the G(V) curve (average slope factors were 13.1 ± 0.2 mV and 17.6 ± 0.5 mV in the absence and presence of 70 μM DHA, respectively). 70 μM DHA did not cause a consistent change in $G_{max}$ of the $hK_V7.4$ channel (average relative $\Delta G_{max}$ was 1.09 ± 0.08, p = 0.28). Because 70 μM DHA induced significant shifts in $V_{50}$ of both $hK_V7.5$ and $hK_V7.4$, although in opposite directions, without affecting $G_{max}$ we will throughout the remainder of this study focus our analysis of PUFA effects on $V_{50}$, and G(V) curves shown in figures reporting on PUFA effects will be normalized to visually emphasize $V_{50}$ shifts.

The DHA-evoked shift in the $V_{50}$ of $hK_V7.5$ was significant at concentrations as low as 7 μM (*Figure 1D*, $\Delta V_{50}$ = –8.5 ± 1.3 mV, p = 0.0002). The concentration-response curve for the DHA effect on $hK_V7.5$ predicts a maximum $V_{50}$ shift of –25.3 mV, with 12 μM required for 50% of the maximum shift ($EC_{50}$ = 12 μM). The DHA effect on the $V_{50}$ of $hK_V7.4$ was also significant at 7 μM (*Figure 1E*, $\Delta V_{50}$ = +4.8 ± 1.6 mV, p = 0.01). The concentration-response curve for the DHA effect on $hK_V7.4$ predicts a maximum $V_{50}$ shift of +13.8 mV, with an $EC_{50}$ of 14 μM. The onset of the DHA effect was relatively fast for both the $hK_V7.5$ and $hK_V7.4$ channels, reaching a stable level within about 6 min of application (*Figure 1—figure supplement 1*). The DHA effect proved difficult to wash out or reverse, as re-perfusion with control solution or control solution supplemented with 100 mg/mL bovine serum albumin (BSA) only partially restored baseline current amplitude for $hK_V7.5$ and $hK_V7.4$ (*Figure 1— figure supplement 1*). Altogether, the PUFA DHA induces a concentration-dependent *facilitation* of activation of the $hK_V7.5$ channel, and a concentration-dependent *inhibition* of activation of the $hK_V7.4$ channel.

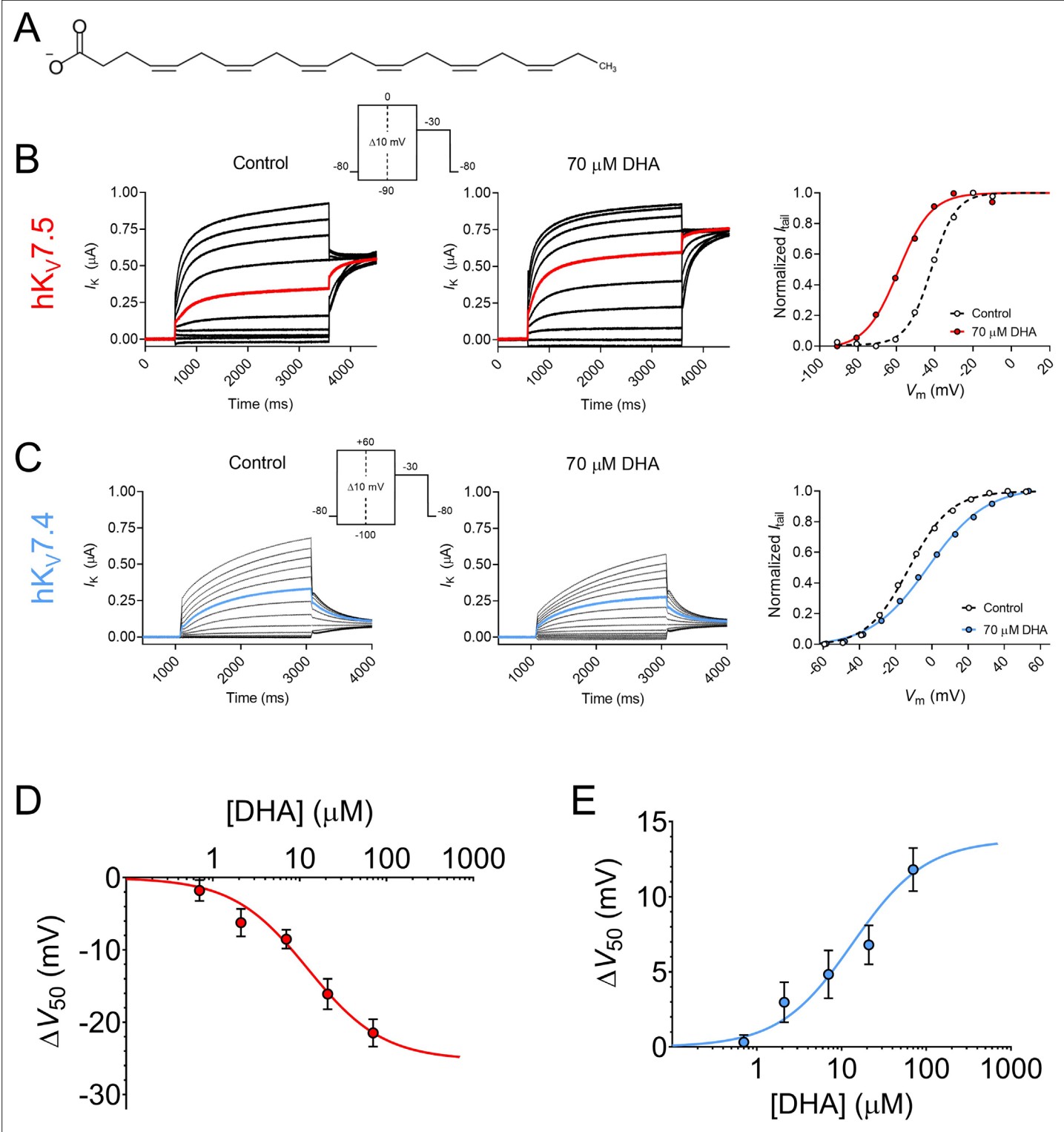

**Figure 1.** Docosahexaenoic acid facilitates the activation of hKv7.5 but inhibits the activation of hKv7.4. (**A**) Molecular structure of DHA. (**B**) Representative current family with corresponding G(V) curve of hKv7.5 in the absence (left) and presence (middle) of 70 μM DHA. Currents generated by the voltage protocol shown as inset. Red traces denote current generated by a test voltage to –40 mV. The G(V) curves (right) have been normalized between 0 and 1, as described in Materials and methods, to better visualize shifts in $V_{50}$. Curves represent Boltzmann fits (see Materials and methods for details). $V_{50}$ for this specific cell: $V_{50,ctrl}$ = –41.7 mV, $V_{50,DHA}$ = –59 mV. (**C**) Same as in B but for hKv7.4. Blue traces denote current generated by a test voltage to 0 mV. $V_{50}$ for this specific cell: $V_{50,ctrl}$ = –12.6 mV, $V_{50,DHA}$ = –2.0 mV. (**D–E**) Concentration-response curve of the DHA effect on $V_{50}$ of hKv7.5 (**D**) and hKv7.4 (**E**). Curves represent concentration-response fits (see Materials and methods for details). Best fits: $\Delta V_{50,max}$ is –25.3 mV for hKv7.5

*Figure 1 continued on next page*

and +13.8 mV for $hK_V7.4$. $EC_{50}$ is 12 µM for $hK_V7.5$ and 14 µM for $hK_V7.4$. Data shown as mean ± SEM. n = 3–15. See also *Figure 1—figure supplement 1*, *Figure 1—source data 1*.

The online version of this article includes the following source data and figure supplement(s) for figure 1:

**Source data 1.** Numerical data for *Figure 1*.

**Figure supplement 1.** The DHA response has a rapid onset for both $hK_V7.5$ and $hK_V7.4$.

## The $hK_V7.5$ response, but not the $hK_V7.4$ response, changes direction in an electrostatic manner

To further understand the molecular basis of the DHA response, we examined the importance of the charge of the DHA head group. Several previous studies identify electrostatic interactions between positively charged arginines in the upper half of S4 of the VSD and the negatively charged head group on PUFAs (and their analogues) as fundamental for facilitating the activation of $hK_V7.1$, $hK_V7.2/7.3$ and some other $K_V$ channels (*Börjesson and Elinder, 2011*; *Liin et al., 2015*; *Liin et al., 2016*; *Liin et al., 2018*; *Martín et al., 2021*). This is seen as a shift in $V_{50}$ toward more negative voltages. The electrostatic PUFA effect on $V_{50}$ can be tuned, from inducing negative to positive shifts in $V_{50}$, by altering the charge of the PUFA head group (*Börjesson et al., 2010*; *Liin et al., 2015*). To investigate if the

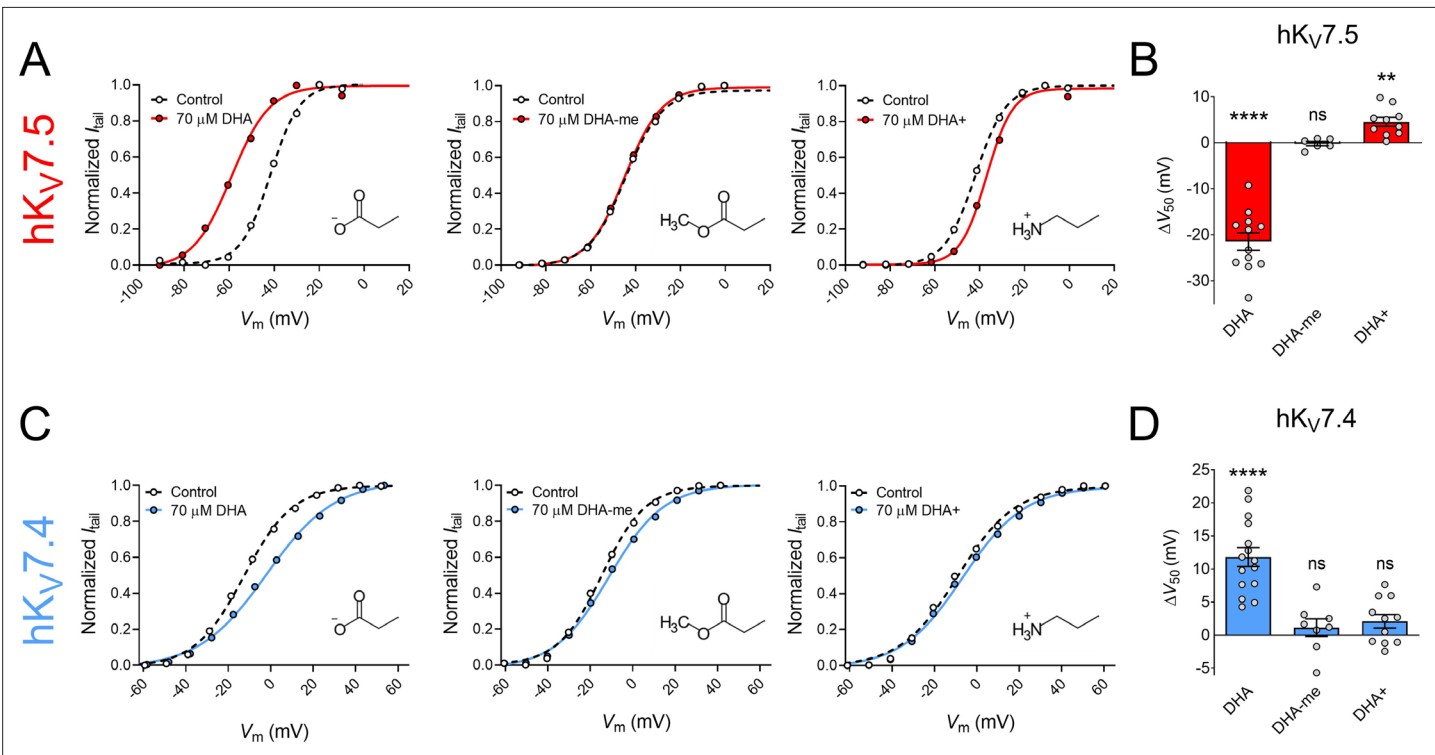

**Figure 2.** The impact of PUFA head group charge for $hK_V7.5$ and $hK_V7.4$ effects. (**A–B**) Impact of the head group charge on the ability of DHA to shift $V_{50}$ of $hK_V7.5$, assessed by comparing the effect of negatively charged DHA, uncharged DHA-me, and positively charged DHA+ at 70 µM. (**A**) Representative examples of the negative shift in $V_{50}$ induced by DHA (left, same data as shown in *Figure 1B*), the lack of effect of DHA-me (middle, $V_{50,ctrl}$ = –43.9 mV; $V_{50,DHA-me}$ = -44.6 mV), and the positive shift in $V_{50}$ induced by DHA+ (right, $V_{50,ctrl}$ = –42.0 mV; $V_{50,DHA+}$ = –37.0 mV). Molecular structure of head groups are shown as insets. (**B**) Summary of responses to indicated DHA compound. Data shown as mean ± SEM. n = 6–12. Statistics denote one sample *t* test against a hypothetical mean of 0 mV. ns denotes not significant, ** denotes p ≤ 0.01, **** denotes p ≤ 0.0001. (**C–D**) same as in A-B but for $hK_V7.4$. (**C**) Representative examples of the depolarizing shift in $V_{50}$ induced by DHA (left, same data as shown in *Figure 1C*), the lack of effect of DHA-me (middle, $V_{50,ctrl}$ = –14.6 mV; $V_{50,DHA-me}$ = –11.1 mV), and the lack of effect of DHA+ (right, $V_{50,ctrl}$ = –9.1 mV; $V_{50,DHA+}$ = –6.6 mV). (**D**) Summary of responses to indicated DHA compound. Data shown as mean ± SEM. n = 8–15. Statistics denote one sample *t* test against a hypothetical mean of 0 mV. ns denotes not significant, **** denotes p ≤ 0.0001. See also *Figure 2—source data 1*.

The online version of this article includes the following source data for figure 2:

**Source data 1.** Numerical data for *Figure 2*.

**Table 1.** List of PUFAs and PUFA analogues used in this study.

| IUPAC name | Lipid no. (C:DBs) | Omega no. | Molecular structure |
|---|---|---|---|
| **PUFAs:** | | | |
| Tetradecatrienoic acid (TTA) | 14:3 | $\omega-3$ | |
| Hexadecatrienoic acid (HTA) | 16:3 | $\omega-3$ | |
| Linoleic acid (LA) | 18:2 | $\omega-6$ | |
| Arachidonic acid (AA) | 20:4 | $\omega-6$ | |
| Eicosapentaenoic acid (EPA) | 20:5 | $\omega-3$ | |
| Docosahexaenoic acid (DHA) | 22:6 | $\omega-3$ | |
| **PUFA analogues:** | | | |
| Arachidonoyl amine (AA+) | 20:4 | $\omega-6$ | |
| Docosahexaneoic acid methyl ester (DHA-me) | 22:6 | $\omega-3$ | |
| Docosahexaenoyl amine (DHA+) | 22:6 | $\omega-3$ | |

C denotes number of carbon atoms in tail, DB denotes double bonds.

same electrostatic mechanism is at play in the responses of $hK_V7.5$ and $hK_V7.4$, we compared the DHA response of the channels with: (1) a DHA analogue with an uncharged methyl ester head group (DHA-me), and (2) a DHA analogue with a positively charged amine head group (DHA+).

70 µM of DHA-me did not significantly shift $V_{50}$ of $hK_V7.5$ (*Figure 2A*, $\Delta V_{50} = -0.2 \pm 0.5$ mV, p = 0.67). In contrast, 70 µM DHA+ brought on a small, but significant positive shift in $V_{50}$ of $hK_V7.5$ (*Figure 2A*, $\Delta V_{50} = +4.5 \pm 0.9$ mV, p = 0.001). Thus, for $hK_V7.5$ the negatively charged DHA facilitates activation, the uncharged DHA-me has no effect, and the positively charged DHA+ inhibits channel activation (effects summarized in bar graph of *Figure 2B*). This is in line with the lipoelectric mechanism that has been proposed to explain PUFA effects on the $hK_V7.1$ and $hK_V7.2/7.3$ channels (*Liin et al., 2015*; *Liin et al., 2016*).

Neither 70 µM of DHA-me nor 70 µM of DHA+ had significant effects on $V_{50}$ of $hK_V7.4$ (*Figure 2C*, $\Delta V_{50} = +1.1 \pm 1.3$ mV, p = 0.42 for DHA-me; $\Delta V_{50} = +2.1 \pm 1.0$ mV, p = 0.068 for DHA+). Thus, while the negatively charged DHA caused a positive shift in $V_{50}$ of $hK_V7.4$, neither the uncharged DHA-me nor the positively charged DHA+ altered the $V_{50}$ of $hK_V7.4$ (effects summarized in bar graph of *Figure 2D*). Even though a negative charge of the head group seems to be important for the effect, the responses are not in line with the lipoelectric mechanism, making $hK_V7.4$ unique among the $hK_V7$ channels.

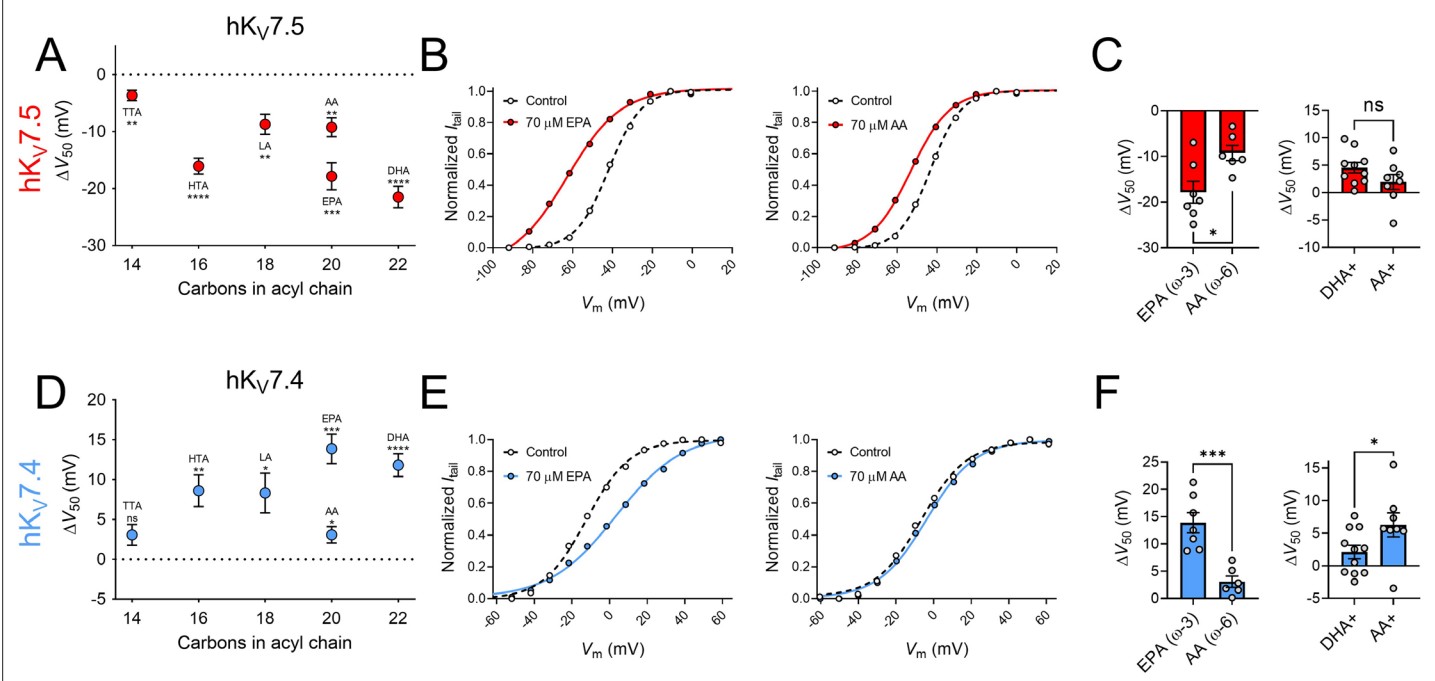

**Figure 3.** The impact of PUFA tail properties for hK$_V$7.5 and hK$_V$7.4 effects. (**A–C**) Impact of PUFA tail properties for the ability of PUFAs to shift V$_{50}$ of hK$_V$7.5, assessed by comparing the response to 70 µM of indicated PUFAs (see **Table 1** for molecular structures). (**A**) Average effect of indicated PUFAs. Data shown as mean ± SEM. n = 6–8. Statistics denote one sample *t* test against a hypothetical mean of 0 mV ns denotes not significant, ** denotes p ≤ 0.01, *** denotes p ≤ 0.001, **** denotes p ≤ 0.0001. (**B**) Representative examples of response to 70 µM of the ω –3 PUFA EPA (left, V$_{50,ctrl}$ = -41.9 mV; V$_{50,EPA}$ = -62.8 mV) and ω –6 PUFA AA (middle, V$_{50,ctrl}$ = -43.3 mV; V$_{50,AA}$ = -53.1 mV). (**C**) Bar graphs with comparison of average response to 70 µM of EPA and AA (n = 6–7) and DHA+ and AA+ (n = 8–10). Statistics denote Student's *t* test. ns denotes not significant, * denotes p ≤ 0.05. (**D–F**) same as in A-C but for hK$_V$7.4. (**D**) n = 6–8. (**E**) Left: V$_{50,ctrl}$ = –12.3 mV; V$_{50,EPA}$ = +3.4 mV. Middle: V$_{50,ctrl}$ = –6.5 mV; V$_{50,AA}$ = –3.6 mV. (**F**) n = 6–7 and n = 8–11, respectively. * denotes p ≤ 0.05, ** denotes p ≤ 0.01, *** denotes p ≤ 0.001, **** denotes p ≤ 0.0001. See also **Figure 3—source data 1**.

The online version of this article includes the following source data for figure 3:

**Source data 1.** Numerical data for **Figure 3**.

## Both hK$_V$7.5 and hK$_V$7.4 respond broadly to PUFAs, although with varied magnitudes of responses

Next, we studied if the effects on hK$_V$7.5 and hK$_V$7.4 are specific to DHA or if they are shared among different PUFAs by testing a series of PUFAs, listed in **Table 1**. These PUFAs vary in their molecular properties, such as the length of the tail and the number and position of double bonds in the tail. **Figure 3A** shows a plot of the magnitude of ΔV$_{50}$ following exposure of hK$_V$7.5 to PUFAs (at 70 µM) against the number of carbon atoms in each respective PUFA tail. All PUFAs, regardless of length (14–22 carbon atoms) significantly shifted V$_{50}$ of hK$_V$7.5 toward more negative voltages, although to different extents. The magnitude of the PUFA-induced ΔV$_{50}$ of hK$_V$7.5 followed a pattern of TTA < LA = AA < HTA ≤ EPA < DHA. One observation is that the ω –6 PUFAs LA and AA induced smaller shifts than the ω –3 PUFAs HTA, EPA and DHA. Both AA and EPA are 20 carbon atoms long. However, while AA is an ω –6 PUFA, EPA is an ω –3 PUFA (**Table 1**). **Figure 3B** shows representative G(V) curves for hK$_V$7.5 that highlight the larger shift in V$_{50}$ evoked by 70 µM of EPA (left) than by 70 µM of AA (right). On average, EPA induced a shift of –17.9 ± 2.4 mV, which was significantly larger than the shift induced by AA (ΔV$_{50}$ = –9.3 ± 1.7 mV; **Figure 3C**; p = 0.015). These results suggest that in PUFAs of equal length, the ω -number has an impact on the magnitude of the PUFA response of hK$_V$7.5. We also compared the positively charged amine analogue DHA+ to an amine analogue of AA (AA+). The positive shift in V$_{50}$ of hK$_V$7.5 by 70 µM AA+ was slightly smaller than that observed with DHA+, and did not differ significantly from a hypothetical shift of 0 mV (**Figure 3C**, AA+ ΔV$_{50}$ = +1.9 ± 1.4 mV, p = 0.2). However, there was also no statistically significant difference between the effects induced by DHA+ and AA+ (p = 0.13), which indicates that the importance of the ω -number for the amine

**Table 2.** Biophysical properties of tested constructs under control conditions.

| Channel | Variant | $V_{50}$ (mV) Mean ± SEM | Slope (mV) Mean ± SEM | n |
|---|---|---|---|---|
| | Wild-type | −11.1 ± 1.4 | 13.1 ± 0.2 | 20 |
| | F182L | −21.0 ± 2.9 | 12.0 ± 0.5 | 10 |
| | R213Q | 4.6 ± 1.3 | 12.2 ± 0.4 | 24 |
| | R216Q | −26 ± 0.8 | 11.4 ± 0.3 | 24 |
| | R219Q | 13.3 ± 1.0 | 12.7 ± 0.2 | 24 |
| $hK_V7.4$ | S273A | −15.9 ± 3.1 | 14.1 ± 1.9 | 15 |
| $hK_V7.5$ | Wild-type | −42.2 ± 0.7 | 8.4 ± 0.2 | 23 |
| $hK_V7.4/7.5$ | Wild-type | −22.9 ± 0.6 | 10.8 ± 0.2 | 18 |
| $hK_V7.4/KCNE4$ | Wild-type | −12.9 ± 2.2 | 12.9 ± 1.3 | 7 |

n denotes number of cells. $V_{50}$ and slope were determined from Boltzmann fits, as described in Materials and methods. See also Table 2—source data 1.

The online version of this article includes the following source data for table 2:

**Source data 1.** Numerical data for *Table 2*.

analogues should be interpreted with caution. Altogether, these experiments show that many PUFAs activate the $hK_V7.5$ channel and suggest that the $hK_V7.5$ channel shows a preference toward $\omega$−3 over $\omega$−6 PUFAs.

*Figure 3D* shows a plot of the magnitude of $\Delta V_{50}$ following exposure to PUFAs (at 70 µM) against the number of carbon atoms in each respective PUFA tail for the $hK_V7.4$ channel. The shortest PUFA, TTA, did not evoke a significant shift in $V_{50}$ ($\Delta V_{50}$ = +3.1 ± 1.3 mV, p = 0.051). All remaining PUFAs (at a concentration of 70 µM), however, significantly shifted $V_{50}$ of $hK_V7.4$ toward positive voltages. The magnitude of the PUFA-induced $\Delta V_{50}$ of $hK_V7.4$ followed a pattern of TTA <AA < LA ≤ HTA < DHA < EPA with no clear pattern in magnitude of effect based on the $\omega$-number of the PUFA tail. Although the $\omega$−3 PUFA EPA induced a larger shift than the $\omega$−6 PUFA AA (*Figure 3E, F*, EPA $\Delta V_{50}$ = +13.9 ± 1.9 mV; AA $\Delta V_{50}$ = +3.1 ± 1.0 mV, p = <0.001), AA+ caused a larger shift in $V_{50}$ compared to DHA+ (*Figure 3F*, DHA+ $\Delta V_{50}$ = +2.1 ± 1.0 mV; AA+ $\Delta V_{50}$ = +6.3 ± 1.8 mV, p = 0.049). Moreover, the $\omega$−6 PUFA LA induced comparable shifts in $V_{50}$ to those of the $\omega$−3 PUFA HTA. Altogether, these experiments show that many PUFAs inhibit the $hK_V7.4$ channel, with no obvious pattern of which PUFAs $hK_V7.4$ shows a preference toward.

## $hK_V7.4/7.5$ co-expression and disease-associated $hK_V7.4$ mutations, but not hKCNE4 co-expression, influence the response to DHA

We next characterized the DHA response of oocytes co-injected with cRNAs encoding $hK_V7.4$ and $hK_V7.5$ (referred to as $hK_V7.4/7.5$) to allow for the potential formation of heteromeric channels containing $hK_V7.4$ and $hK_V7.5$ subunits. Oocytes co-injected with $hK_V7.4$ and $hK_V7.5$ generated currents with biophysical properties that fell between those of each homomeric channel complex ($hK_V7.4$ $V_{50}$ = −11.1 ± 1.4 mV; $hK_V7.5$ $V_{50}$ = −42.2 ± 0.7 mV; $hK_V7.4/7.5$ $V_{50}$ = −22.9 ± 0.6 mV, *Table 2*). This is in agreement with previous studies showing a $V_{50}$ of co-expressed $hK_V7.4/7.5$ channels intermediate to that of homomeric channels (*Brueggemann et al., 2011*). In addition, 70 µM of DHA induced an effect on $hK_V7.4/7.5$ intermediate to that of homomeric $hK_V7.4$ and $hK_V7.5$ channels, with no change in $V_{50}$ (*Figure 4A, C*, $\Delta V_{50}$ = −0.4 ± 0.6 mV, p = 0.53).

$K_V7.4$ has been shown to co-assemble with the KCNE4 subunit in, for instance, vascular smooth muscle tissue (*Jepps et al., 2015*). We therefore characterized the DHA response of oocytes co-injected with cRNAs encoding $hK_V7.4$ and hKCNE4 (referred to as $hK_V7.4/KCNE4$). In agreement with previous studies (*Vanoye et al., 2009*; *Jepps et al., 2015*), this generated currents with biophysical properties fairly similar to those of $hK_V7.4$ alone (*Table 2*). 70 µM of DHA shifted $V_{50}$ of $hK_V7.4/KCNE4$ toward more positive voltages (*Figure 4B–C*, $\Delta V_{50}$ = +8.9 ± 1.3 mV, p = 0.0005). This effect was

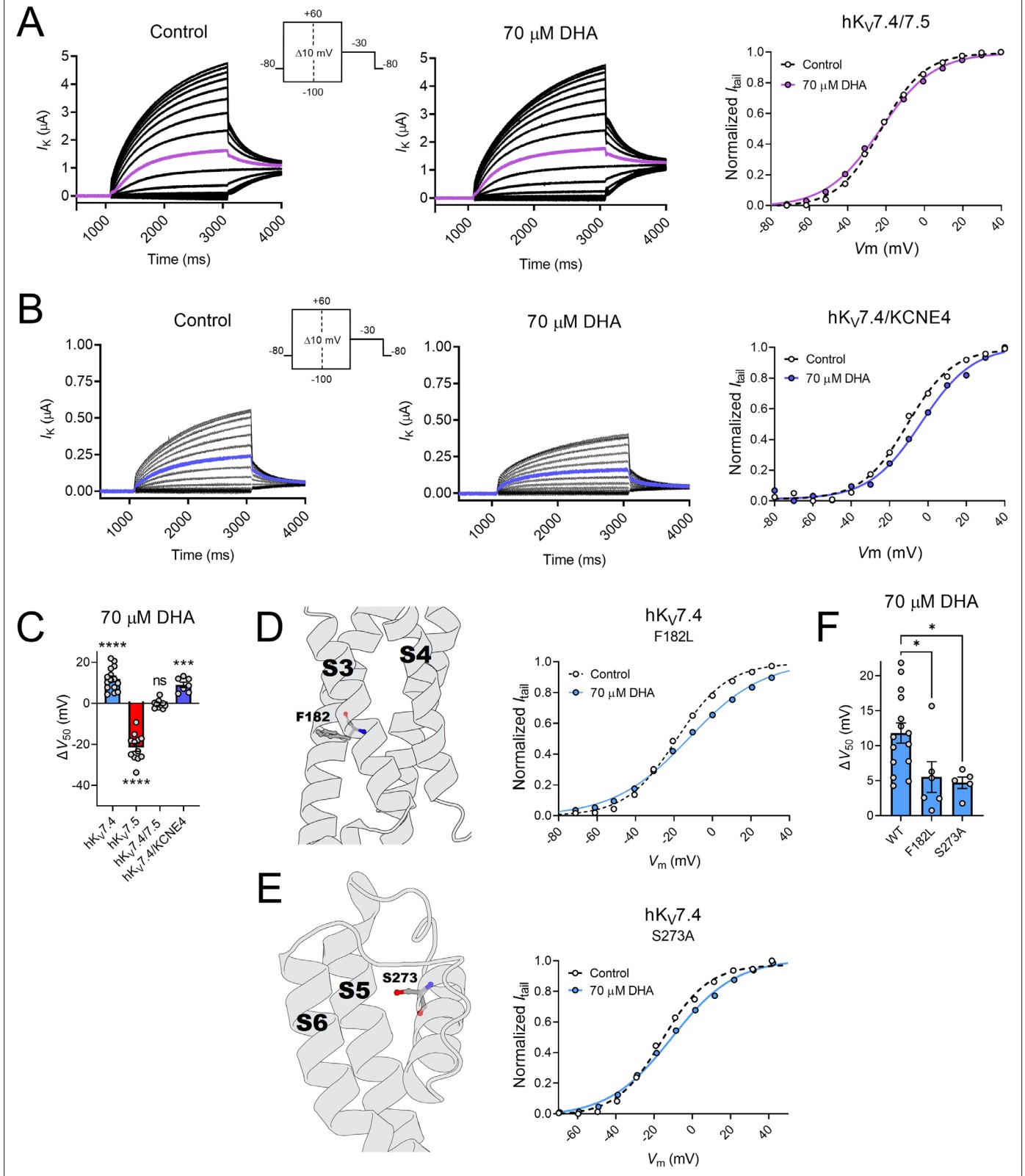

**Figure 4.** hK$_V$7.4/7.5 co-expression and disease-associated hKv7.4 mutations, but not hKCNE4 co-expression, influence the response to DHA. (**A**) Representative current family with corresponding G(V) curve of hK$_V$7.4/7.5 in the absence (left) and presence (middle) of 70 µM DHA. Purple traces denote current generated by a test voltage to –20 mV. Curves represent Boltzmann fits. V$_{50}$ for this specific cell: V$_{50,ctrl}$ = –22.8 mV; V$_{50,DHA}$ = –23.1 mV. (**B**) Representative current family with corresponding G(V) curve of hK$_V$7.4/KCNE4 in the absence (left) and presence (middle) of 70 µM DHA. Blue

*Figure 4 continued*

traces denote current generated by a test voltage to 0 mV. Curves represent Boltzmann fits. $V_{50}$ for this specific cell: $V_{50,ctrl}$ = –10.7 mV; $V_{50,DHA}$ = –3.5 mV. (**C**) Summary of response of hK$_V$7.4/7.5 and hK$_V$7.4/KCNE4 to 70 µM DHA, with responses of hK$_V$7.4 and hK$_V$7.5 for reference. Data shown as mean ± SEM. n = 7–15. Statistics denote one sample *t* test against a hypothetical mean of 0 mV. ns denotes not significant, *** denotes p ≤ 0.001, **** denotes p ≤ 0.0001. (**D–E**) Impact of hK$_V$7.4 mutations F182L (**D**) and S273A (**E**) on the response to DHA. Structural model (PDB ID - 7BYL; *Li et al., 2021*) of hK$_V$7.4 with position of F182 and S273 marked. Representative G(V) curve of indicated hK$_V$7.4 mutants in the absence and presence of 70 µM DHA. Curves represent Boltzmann fits. For these specific cells: F182L: $V_{50,ctrl}$ = –18.7 mV; $V_{50,DHA}$ = –12.4 mV. For S273A: $V_{50,ctrl}$ = –15.8 mV; $V_{50,DHA}$ = –10.8 mV. (**F**) Summary of response of WT hK$_V$7.4 and indicated mutants to 70 µM DHA. Data shown as mean ± SEM. n = 5–15. Statistics denote one-way ANOVA followed by Dunnett's multiple comparisons test to compare the response of mutants to that of the wild-type. * denotes p ≤ 0.05. See also *Figure 4—source data 1*.

The online version of this article includes the following source data for figure 4:

**Source data 1.** Numerical data for *Figure 4*.

comparable to that on hK$_V$7.4 alone (p = 0.23), suggesting the hKCNE4 subunit does not alter the DHA response of the hK$_V$7.4 channel.

Several mutations in the *KCNQ4* gene (encoding hK$_V$7.4) have been identified in humans and are often linked to DFNA2 non-syndromic hearing loss (*Jung et al., 2019*; *Rim et al., 2021*), although the possible contribution to pathology remains to be determined for many mutations. Notably, the F182L and S273A missense mutations in hK$_V$7.4 (*Figure 4D–E*), which are suspected to be linked to impaired hearing (*Jung et al., 2019*; *Kim et al., 2011*), exchanges the native hK$_V$7.4 residue for the hK$_V$7.5 counterpart. By means of site-directed mutagenesis, we substituted these residues in hK$_V$7.4, one by one. Under control conditions, we found both mutants to behave fairly similar to wild-type hK$_V$7.4 (*Table 2*). The S273A mutant had a $V_{50}$ comparable to wild-type hK$_V$7.4, whereas the $V_{50}$ of the F182L mutant was shifted about 10 mV toward more negative voltages compared to wild-type hK$_V$7.4 (*Table 2*). However, both mutations impaired the inhibitory hK$_V$7.4 response to DHA. 70 µM of

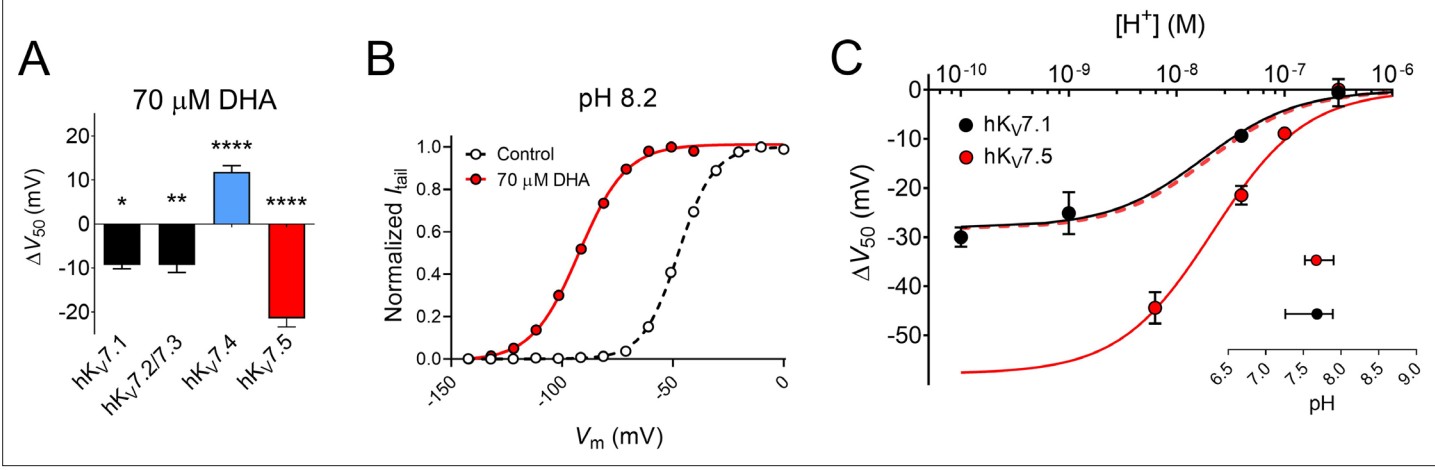

**Figure 5.** Extracellular pH tunes the DHA response of hK$_V$7.5. (**A**) Comparison of the $V_{50}$ shift of hK$_V$7 subtypes in response to 70 µM DHA. Data for hK$_V$7.1 and hK$_V$7.2/7.3 shown as reported in *Liin et al., 2016*; *Liin et al., 2015*. Data for hK$_V$7.4 and hK$_V$7.5 from the present study. Data shown as mean ± SEM. n = 3–15. Statistics denote one sample *t* test against a hypothetical mean of 0 mV. * denotes p ≤ 0.05, ** denotes p ≤ 0.01, **** denotes p ≤ 0.0001. (**B–C**) Altering extracellular pH tunes the shift in $V_{50}$ of hK$_V$7.5 induced by 70 µM DHA, with a greater response observed at more alkaline pH. Note that in all experiments, the pH of the control solution and DHA-supplemented control solution was identical. (**B**) Representative example of the DHA response at pH 8.2 ($V_{50,ctrl}$ = –47.6 mV; $V_{50,DHA}$ = –92.4 mV). (**C**) pH-response curve for the DHA effect on $V_{50}$. Data shown as mean ± SEM. n = 3–9. Curves represent pH-response fits (see Materials and methods for details). Best fit for hK$_V$7.5: $\Delta V_{50,max}$ is –57.9 mV. Data for hK$_V$7.1 as reported in *Liin et al., 2015* is included for comparison (Best fit for hK$_V$7.1: $\Delta V_{50,max}$ is –26.3 mV.) Dashed line denotes the pH-response curve for hK$_V$7.5 normalized to $\Delta V_{50,max}$ for hK$_V$7.1 to illustrate the comparable apparent pKa. Inset shows apparent pKa with 95% CI: hK$_V$7.1 apparent pKa = 7.68 with 95% CI [7.26, 7.89]; hK$_V$7.5 apparent pKa = 7.67 with 95% CI [7.52, 7.90]. Note that inconsistent behavior of hK$_V$7.5 under control conditions at pH higher than 8.2 prevented us from determining the DHA effect on hK$_V$7.5 at pH 9 and 10. See also *Figure 5—figure supplement 1*, *Figure 5—source data 1*.

The online version of this article includes the following source data and figure supplement(s) for figure 5:

**Source data 1.** Numerical data for *Figure 5*.

**Figure supplement 1.** pH dependence of the DHA response of hK$_V$7.5.

DHA shifted $V_{50}$ of the F182L mutant by only +5.5 ± 2.2 mV and the S273A mutant by +4.7 ± 0.8 mV (*Figure 4D–F*).

Altogether, these experiments suggest that co-expression of hK$_V$7.4 and hK$_V$7.5 subunits, or substitution of specific residues in hK$_V$7.4 to the hK$_V$7.5 counterpart, alter the DHA response of hK$_V$7.4 to approach that of hK$_V$7.5. However, hK$_V$7.4 co-expression with the hKCNE4 auxiliary subunit did not alter the DHA response.

## The DHA response of hK$_V$7.5 is greater than that of other hK$_V$7 subtypes

We have in previous studies shown that 70 μM of DHA shifts the $V_{50}$ of the hK$_V$7.1 and hK$_V$7.2/7.3 channels by about –9 mV (*Figure 5A*; hK$_V$7.1 $\Delta V_{50}$ = –9.3 ± 0.9 mV, as reported in *Liin et al., 2015*; hK$_V$7.2/7.3 $\Delta V_{50}$ = –9.3 ± 1.7 mV, as reported in *Liin et al., 2016*). Thus, the extent of the hK$_V$7.5 shift induced by 70 μM DHA is greater than that of hK$_V$7.1 or hK$_V$7.2/7.3 (*Figure 5A*). Our experiments indicated the necessity of a negatively charged DHA head group to elicit the activating effect on hK$_V$7.5 (see *Figure 2A*). A negatively charged head group is promoted by alkaline pH, which triggers proton dissociation (*Börjesson et al., 2008*; *Hamilton, 1998*). In a previous study, the apparent pKa of DHA when near hK$_V$7.1 (i.e. the pH at which 50% of the maximal DHA effect is seen, interpreted as the pH at which 50% of the DHA molecules in a lipid environment are negatively charged) was determined to be pH 7.7 (*Liin et al., 2015*). One possible underlying cause of the larger DHA effect on hK$_V$7.5 is that the local pH environment at hK$_V$7.5 promotes DHA deprotonation (i.e. inducing a lower apparent pKa of DHA), thus rendering a greater fraction of the DHA molecules negatively charged and capable of activating the channel. To test this, we assessed the effect of 70 μM DHA on hK$_V$7.5 with the extracellular pH adjusted to either more alkaline (pH = 8.2) or acidic (pH = 6.5, or 7.0) values. Adjusting the extracellular pH had only minor effects on $V_{50}$ under control conditions (*Figure 5—figure supplement 1*). In contrast, the magnitude of the DHA-induced shift varied with pH. At pH 8.2, at which a majority of DHA molecules are expected to be deprotonated, DHA shifts of $V_{50}$ were almost two-fold greater than at physiological pH (pH 7.4 $\Delta V_{50}$ = –21.5 ± 1.9 mV; pH 8.2 $\Delta V_{50}$ = –44.4 ± 3.2 mV, p < 0.0001, Student's *t* test, representative example in *Figure 5B*). The magnitude of the DHA effect was reduced as the pH was gradually titrated toward more acidic pH (*Figure 5C*; *Figure 5—figure supplement 1*), with no shift in $V_{50}$ by 70 μM DHA observed at pH 6.5 ($\Delta V_{50}$ = +0.1 ± 1.0 mV, p = 0.96). The pH dependence of $\Delta V_{50}$ induced by DHA for the hK$_V$7.1 channel is also plotted in *Figure 5C*, to allow for comparison between the two hK$_V$7 subtypes. While there is a clear difference in the extrapolated maximum shifts for the two channels (hK$_V$7.1 $\Delta V_{50,max}$ = –26.3 mV with 95% CI [-18.4,–34.3]; hK$_V$7.5 $\Delta V_{50,max}$ = –57.9 mV with 95% CI [-47.6,–68.2]), the pH required to induce 50% of the maximum $\Delta V_{50}$ (i.e. the apparent pKa values) are similar (inset of *Figure 5C*, hK$_V$7.1 apparent pKa = 7.68 with 95% CI [7.26, 7.89]; hK$_V$7.5 apparent pKa = 7.67 with 95% CI [7.52, 7.90]). This indicates that the difference in the DHA effect between hK$_V$7.1 and hK$_V$7.5 is a question of increased magnitude, rather than a matter of different apparent pKa of DHA at the channels.

## Molecular dynamics simulations predict an unconventional inner VSD PUFA site in hK$_V$7.4

We next turned our attention toward structural components of the hK$_V$7.4 channel that may contribute to its unusual PUFA response. In order to determine the interaction sites for DHA on hK$_V$7.4, we performed coarse-grained molecular dynamics simulations of the channel, solved in an intermediate activation state, embedded in a membrane consisting of 75% POPC and 25% negatively charged DHA randomly distributed in both membrane leaflets (*Figure 6A*). Analysis of the residence time of DHA showed overall prolonged interactions with residues on S1 (L91, Y101, and I105), on S2 (V142, F143, and D146), on S4 (R213) and at the top of the pore region (*Figure 6B* and *Figure 6—figure supplement 1A*). Out of the 13 interaction sites characterized by density-based clustering (*Figure 6—figure supplement 1B*), the site at which DHA interacted with the channel the longest (2.156 μs, cluster #3) was comprised of residues located on the lower half of S4. This site was also occupied ~87% of the time (*Figure 6C*, *Figure 6—figure supplement 1B*), suggesting favorable interactions between the side chains of R213, R216, and R219 and the head group of DHA, as well as interactions with the backbone of M217. In the representative binding pose of that site, the head group of DHA is bound in the lumen of the lower portion of the VSD, close to the S4 helix (*Figure 6D*). We will refer to this as an

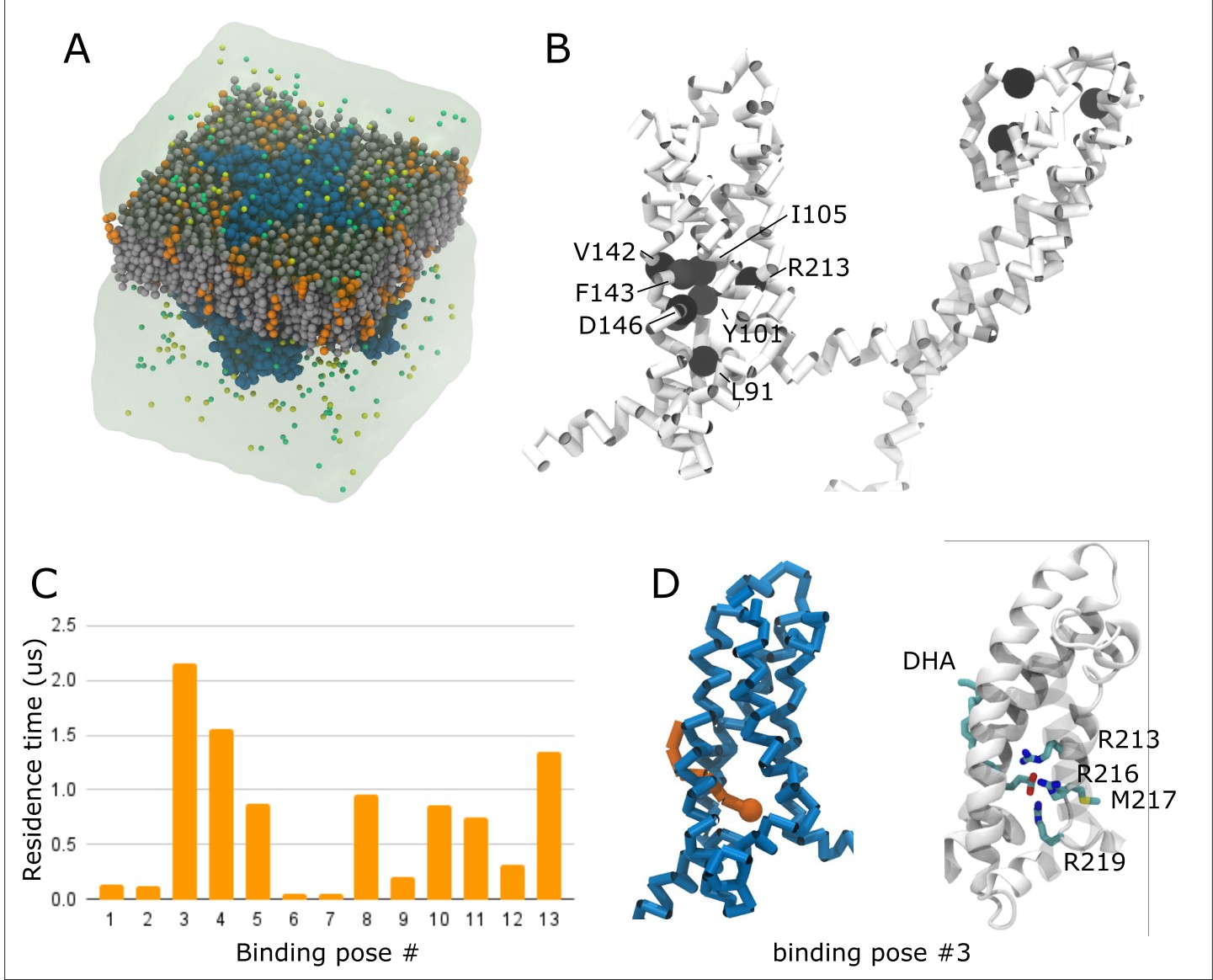

**Figure 6.** Molecular dynamics simulation predicts an unconventional inner VSD site for DHA in hK$_V$7.4. (**A**) Coarse-grained simulation system, containing hK$_V$7.4 (blue) in a bilayer made of 75% POPC (grey) and 25% DHA (orange) plunged in a KCl bath. (**B**) Residence times of interactions between DHA and individual hK$_V$7.4 residues projected as a color scale ranging from white (short residence times) to black (long residence times). Residues with a residence time above 4.4 µs are highlighted as black spheres. Note the cluster of such residues in the lower half of the VSD. (**C**) Residence time of DHA in 13 different binding poses identified through a clustering analysis. (**D**) The binding pose in the cluster with the highest residence time (cluster #3) is shown in the coarse-grained representation used to run the simulations (left - hK$_V$7.4 VSD in blue, DHA in orange) and in an all-atom representation obtained by backmapping the coarse-grained system to all-atom representation (right – protein backbone in white ribbons, DHA and residues interacting with the charged head group as sticks colored by atom type, C – cyan, N – blue, O – red, S – yellow, H omitted for clarity). See also *Figure 6—figure supplements 1–3*. Source data, including code, for molecular dynamics simulations is accessible at https://osf.io/6fuqs/.

The online version of this article includes the following figure supplement(s) for figure 6:

**Figure supplement 1.** Overview of predicted interaction of negatively charged DHA with hK$_V$7.4.

**Figure supplement 2.** Overview of predicted interaction of uncharged DHA with hK$_V$7.4.

**Figure supplement 3.** Structural alignment between the VSDs of hK$_V$7.1 and hK$_V$7.4 highlighting differences in the inner VSD site.

unconventional 'inner VSD site', in contrast to the conventional 'outer VSD site' described previously for K$_V$7.1 (*Yazdi et al., 2021*). Because of its negative charge, DHA in the inner VSD site presumably prevents the outward motion of the positively charged S4 helix, rationalizing its inhibitory effect. The wedging of the tail between S1 and S2 suggests that the DHA head group accesses this interaction

site via this lipid-exposed interface. Repeating these simulations with uncharged DHA changes the interaction pattern drastically. Indeed, neutral DHA shows a high propensity to localize close to the lower half of the S5 and S6 helix, at a considerable distance from the VSD (*Figure 6—figure supplement 2*). This site presumably has little functional effect, providing an explanation for the lack of effect of the uncharged DHA-me on $hK_V7.4$ activation.

## Arginine residues in the lower half of S4 of $hK_V7.4$ contribute to the PUFA response

The molecular dynamics simulations were performed using a $hK_V7.4$ model based on the cryo-EM structure of $hK_V7.4$ (PDB ID - 7BYL), in which S4 is presumably in an intermediate state (*Li et al., 2021*). S4 is anticipated to move further downward when adopting a resting state and further upward when adopting an activated state. The lack of $hK_V7.4$ structure with S4 in a resting state prevents us from using simulations to assess DHA interaction with the inner VSD site in this conformational state. Therefore, to functionally assess the importance of the interactions predicted by the molecular dynamics simulations, we substituted each of the innermost arginines (R213, R216, and R219), one by one, with an uncharged glutamine. This generated mutant $hK_V7.4$ channels with $V_{50}$ values shifted toward more positive (R213Q and R219Q) or negative (R216Q) voltages relative to the wild-type channel (*Table 2*). However, in contrast to the positive shift in $V_{50}$ normally seen in wild-type $hK_V7.4$ by DHA, each of these arginine mutants responded to 70 µM DHA with a negative shift in $V_{50}$ (*Figure 7A–D*, R213Q $\Delta V_{50} = -8.8 \pm 0.9$ mV; R216Q $\Delta V_{50} = -5.7 \pm 0.6$ mV; R219Q $\Delta V_{50} = -7.3 \pm 0.6$ mV). Thus, neutralization of any of these three arginines in the lower half of S4 endowed the $hK_V7.4$ channel with a DHA response typical of other $hK_V7$ channels (that is, PUFA-mediated facilitation of channel activation), confirming the interaction site predictions from the molecular dynamics simulations. With this typical DHA response, the arginine mutants showed a response more in line with the lipoelectric mechanism, with a negative shift in $V_{50}$ induced by negatively charged PUFAs, no or minimal shift in $V_{50}$ induced by uncharged PUFA analogues, and a positive shift in $V_{50}$ induced by positively charged PUFA analogues (*Figure 7—figure supplement 1*). This response pattern was clearest for R213Q, which showed the largest negative shift induced by DHA, for which DHA-me had minimal effects on $V_{50}$ ($\Delta V_{50} = +0.3 \pm 0.1$ mV, p < 0.05) whereas DHA+ and AA+ shifted $V_{50}$ toward positive voltages ($\Delta V_{50} = +2.6 \pm 0.5$ mV, p < 0.005 for DHA+; $\Delta V_{50} = +3.1 \pm 0.4$ mV, p < 0.001 for AA+). The response pattern showed a similar trend for R216Q and R219Q, however, with less robust effects induced by the positively charged compounds (For R216Q: $\Delta V_{50} = +1.6 \pm 0.8$ mV, p = 0.11 for DHA+; $\Delta V_{50} = +5.6 \pm 2.4$ mV, p = 0.06 for AA+; for R219Q: $\Delta V_{50} = +2.2 \pm 1.0$ mV, p = 0.08 for DHA+; $\Delta V_{50} = +4.7 \pm 0.7$ mV, p < 0.005 for AA+). The positive shifts induced by AA+ on the arginine mutants were larger than for $hK_V7.5$ (*Figure 3C*) and those previously reported for the Shaker $K_V$ channel (*Börjesson et al., 2010*), comparable to the effect on WT $hK_V7.4$ (*Figure 3F*), and slightly smaller than those previously reported for $hK_V7.1$ (*Liin et al., 2015*).

## Discussion

This study finds that the activity of $hK_V7.4$ and $hK_V7.5$ channels is modulated by PUFAs. As such, our study expands upon the understanding of the modulation of the $hK_V7$ family of ion channels by this class of lipids. Importantly, we find that both $hK_V7.4$ and $hK_V7.5$ show surprising responses to PUFAs, compared to previously reported effects on other $hK_V7$ subtypes. $hK_V7.4$ is the only $hK_V7$ subtype for which PUFAs induce channel inhibition, whereas $\omega-3$ PUFAs induce unexpectedly large $hK_V7.5$ channel activation. Experiments on co-expressed $hK_V7.4/7.5$ channels and $hK_V7.4$ channels carrying $hK_V7.5$ mimetic disease-associated mutations indicate responses to PUFAs that are intermediate of the $hK_V7.4$ and $hK_V7.5$ homomers. In contrast, hKCNE4 co-expression did not alter the PUFA response of $hK_V7.4$. The diverse responses of $hK_V7$ subtypes to PUFAs demonstrate the importance of carrying out functional experiments on each of the channel subtypes to allow for an understanding of subtype variability in channel modulation.

What mechanistic basis may underlie PUFA-induced facilitation of $hK_V7.5$ activation? We find that the pattern of how $hK_V7.5$ responds to PUFAs conforms to the lipoelectric mechanism that has previously been described for the $hK_V7.1$ and $hK_V7.2/7.3$ channels, with a hyperpolarizing shift in $V_{50}$ induced by negatively charged PUFAs, a depolarizing shift in $V_{50}$ induced by a positively charged

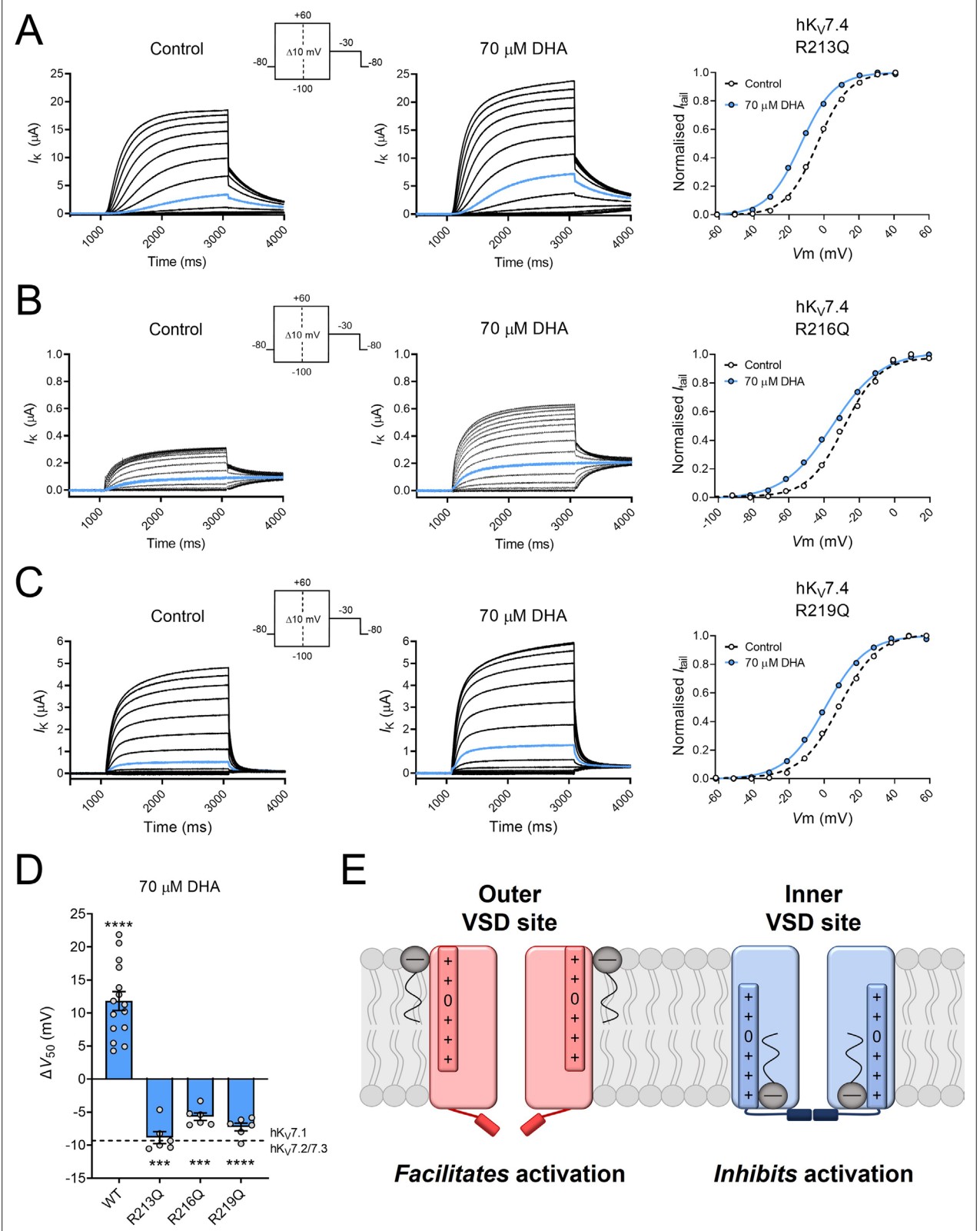

**Figure 7.** Substitution of innermost S4 arginines with charge-neutral glutamine unmasks PUFA-mediated facilitation of hK$_V$7.4 activation.
(**A**) Representative current family with corresponding G(V) curve of hK$_V$7.4 R213Q in the absence (left) and presence (middle) of 70 µM DHA. Blue traces denote current generated by a test voltage to –10 mV. Curves represent Boltzmann fits. V$_{50}$ for this specific cell: V$_{50,ctrl}$ = –3.8 mV; V$_{50,DHA}$ = –13.2 mV.
(**B**) Same as in A but for hK$_V$7.4 R216Q. Blue traces denote current generated by a test voltage to –30 mV. V$_{50}$ for this specific cell: V$_{50,ctrl}$ = -28.7 mV; V$_{50,DHA}$

*Figure 7 continued on next page*

*Figure 7 continued*

= –35.3 mV. (**C**) Same as in A but for hK$_V$7.4 R219Q. Blue traces denote current generated by a test voltage to –10 mV. V$_{50}$ for this specific cell: V$_{50,ctrl}$ = 8.6 mV; V$_{50,DHA}$ = +0.6 mV. (**D**) Summary of the response of hK$_V$7.4 gating charge mutants to 70 μM DHA. Data for wild-type hK$_V$7.4 has been added for reference. Dashed line indicates the mean ΔV$_{50}$ observed in hK$_V$7.1 and hK$_V$7.2/7.3 channels exposed to 70 μM DHA (same data as in *Figure 5A*). Data shown as mean ± SEM. n = 6–15. Statistics denote one sample *t* test against a hypothetical mean of 0 mV. *** denotes p ≤ 0.001, **** denotes p ≤ 0.0001. (**E**) Cartoon illustrating facilitation of channel opening from the conventional outer VSD site (*left*, in red) and inhibition of channel opening from the unconventional inner VSD site (*right*, in blue) mediated by PUFAs (gray). In the outer VSD site, PUFAs interact with the outermost positively charged gating charges of S4, which facilitates channel activation and channel opening. In the inner VSD site, PUFAs interact with the innermost gating charges of S4, stabilizing an intermediate or resting conformation that impairs channel opening. See also *Figure 7—figure supplement 1*, *Figure 7—source data 1*.

The online version of this article includes the following source data and figure supplement(s) for figure 7:

**Source data 1.** Numerical data for *Figure 7*.

**Figure supplement 1.** The impact of PUFA head group charge for effects on hK$_V$7.4 arginine mutants.

PUFA analogue, and no effect of an uncharged PUFA analogue. Therefore, we find it likely that PUFAs also facilitate activation of hK$_V$7.5 through electrostatic interactions with positively charged gating charges in the upper half of the VSD (schematically illustrated in *Figure 7E left*). However, $\omega$–6 PUFAs are less effective than $\omega$–3 PUFAs at activating the hK$_V$7.5 channel. This is different from the hK$_V$7.1/ KCNE1 channel, for which $\omega$–6 or $\omega$–9 PUFAs were identified as the best modulators of channel activity (*Bohannon et al., 2019*). Although the role of the $\omega$-numbering in determining the extent of PUFA effects on hK$_V$7.5 should be interpreted with caution (for instance, because the PUFAs also vary in their number of double bonds, see *Table 1*), it is clear that hK$_V$7.5 and hK$_V$7.1/KCNE1 do not follow the same response pattern to different PUFAs. Work on hK$_V$7.1/KCNE1 has proposed that the KCNE1 subunit impairs the DHA response of hK$_V$7.1 by rearranging one of the external loops in hK$_V$7.1, which alters the pH at the outer VSD site of the channel to promote DHA protonation (*Larsson et al., 2018*; *Liin et al., 2015*). This led us to test if the greater DHA response of the hK$_V$7.5 channel was caused by a lower apparent pKa of DHA at hK$_V$7.5 compared to hK$_V$7.1–7.3, which might promote DHA deprotonation. However, the similar apparent pKa values of DHA for hK$_V$7.1 and hK$_V$7.5 discards this hypothesis. Instead, we find a larger magnitude of the DHA effect on hK$_V$7.5 at all pH values compared to hK$_V$7.1. Interestingly, a residue at the top of S5 of hK$_V$7.1, Y278, which was recently identified as an important 'anchor point' for binding the $\omega$–6 PUFA LA near the VSD to allow for efficient shifts in V$_{50}$ (*Yazdi et al., 2021*), is instead a phenylalanine (F282) in hK$_V$7.5. A phenylalanine mutation of this 'anchor point' in hK$_V$7.1 (Y278F) led to a drastic decrease in apparent affinity of LA for hK$_V$7.1, possibly by removing hydrogen bonding between the head group of LA and the hydroxyl group of the tyrosine side chain (*Yazdi et al., 2021*). However, because the Y278F mutation of hK$_V$7.1 also reduced the apparent affinity of DHA for hK$_V$7.1 (*Yazdi et al., 2021*), and DHA was the most potent of the PUFAs we tested on hK$_V$7.5 in this study, sequence variability at this specific position is not likely to underlie the relatively larger effect of DHA on hK$_V$7.5 compared to hK$_V$7.1. Thus, we conclude the nature of PUFA binding to hK$_V$7 channels is more complex and there may be other stabilizing residues in the PUFA interaction site of the hK$_V$7.5 channel.

We find that the activity of the hK$_V$7.4 channel is inhibited by PUFAs, and that the direction of the effect is not reversed by reversing the charge of the PUFA head group. Therefore, the PUFA effect on hK$_V$7.4 does not conform to the lipoelectric mechanism proposed for other K$_V$7 channels. Additionally, we did not find any relationship between PUFA tail properties and the response of hK$_V$7.4 to PUFAs, other than that all PUFAs with significant effects shifted V$_{50}$ toward more positive voltages. A cryo-EM structure for hK$_V$7.4 was recently reported (PDB ID – 7BYL; *Li et al., 2021*), which reveals structural differences when comparing to the hK$_V$7.1 structure (PDB ID – 6UZZ; *Sun and MacKinnon, 2020*). For instance, the entire VSD of hK$_V$7.4 is rotated 15° clockwise relative to the pore domain of the channel. In molecular dynamics simulations, we also identified an unconventional predicted DHA site in hK$_V$7.4. Although the lack of hK$_V$7.4 structures in different conformational states limits the ability to simulate the state dependence of DHA interaction with different sites, our experiments validated the functional relevance of the unconventional predicted site. Taken together, our simulation and experimental data suggest that hK$_V$7.4 is unique among hK$_V$7 channels in harboring a functional PUFA site in the inner half of the VSD. The DHA-induced shift in V$_{50}$ toward positive voltages in wild-type hK$_V$7.4 is compatible with DHA stabilizing a resting or intermediate state of S4 through interaction

with arginines in the lower half of S4 (schematically illustrated in *Figure 7E right*). Disruption of DHA interaction with this inner VSD site, through substitution of the innermost S4 arginines, endowed hK$_V$7.4 with the typical PUFA response of a shift in V$_{50}$ toward negative voltages. We therefore suggest that hK$_V$7.4 also harbors a conventional PUFA site at the outer half of the S4, the functional effect of which is unmasked upon disruption of the functionally dominant inner VSD site. In agreement with this, weak DHA interaction with the conventional outer VSD site is indeed observed in our simulations (corresponding to cluster #9, see *Figure 6—figure supplement 1B*). Furthermore, our experiments using PUFA analogues indicate that it is possible to charge-tune the PUFA response of the hK$_V$7.4 arginine mutants from a negative to positive shift in V$_{50}$ by altering the charge of the PUFA head group, a feature which is in line with observations on other K$_V$ channels electrostatically modulated by PUFAs (*Börjesson et al., 2010*; *Liin et al., 2015*). A structural alignment between the VSDs of hK$_V$7.1 (PDB ID – 6UZZ) and hK$_V$7.4 (PDB ID – 7BYL) revealed two possible molecular level reasons for the differential binding of DHA to these two channels: in hK$_V$7.4 the binding site we uncovered indeed appears both larger, with the S4 helix pushed inwards in hK$_V$7.1 relative to hK$_V$7.4, and more accessible, as the cleft between S1 and S2 via which DHA appears to enter the cavity features a bulky, obstructing phenylalanine residue (F166) in hK$_V$7.1 compared to a smaller valine residue (V142) in hK$_V$7.4 (*Figure 6—figure supplement 3*).

What potential physiological implications may our findings have? Increased consumption of foods rich in PUFAs has long been associated with improved cardiovascular health (*Bang et al., 1980*; *Kagawa et al., 1982*; *Saravanan et al., 2010*) and multiple mechanisms have been proposed, including those mediated by the immune system, the endothelium and vascular smooth muscle tissue (*Massaro et al., 2008*). Several ion channels in both endothelial cells and VSMCs have been studied as the molecular correlates of PUFA-mediated vasodilatory mechanisms (*Bercea et al., 2021*). Of note, PUFA-induced inhibition of L-type Ca$_V$ channels (*Engler et al., 2000*) and PUFA-induced activation of large conductance Ca$^{2+}$-activated K$^+$ (BK) channels contribute to the vasodilation induced by PUFAs (*Hoshi et al., 2013*; *Limbu et al., 2018*). Intriguingly, a recent study by *Limbu et al., 2018* observed an endothelium-independent residual relaxation of rodent arteries induced by $\omega$−3 PUFAs despite pre-treatment with several K$^+$ channel inhibitors, suggesting there may be another mechanism underlying PUFA-mediated vasorelaxation that does not involve BK channels. Our finding that PUFAs facilitate activation of hK$_V$7.5 raises the possibility that K$_V$7.5 subunits contribute to the residual vasorelaxation observed by Limbu and colleagues.

Besides cardiovascular effects, an increased intake of $\omega$−3 PUFAs has also indicated a potential protective role against hearing loss (*Curhan et al., 2014*; *Dullemeijer et al., 2010*; *Gopinath et al., 2010*). While these studies propose the protective mechanism of PUFAs on hearing stem from vascular effects that improve cochlear perfusion, no direct mechanism of PUFAs on the hearing organ was investigated. Based on our findings, it would be unlikely that the decreased risk of hearing loss is due to direct actions of PUFAs on K$_V$7.4 channels expressed in OHCs, given that we find the channel activity to be inhibited by PUFAs. However, PUFA-induced facilitation of the activity of other K$_V$7 channels (e.g. K$_V$7.1 and K$_V$7.5) in VSMCs could possibly contribute to improved cochlear perfusion. Of note, the two DFNA2-associated missense mutations in hK$_V$7.4 we examined, F182L and S273A, showed intrinsic biophysical behavior comparable to that of wild-type hK$_V$7.4 (see *Table 2*). This is in overall agreement with previous studies of these mutants performed in other expression systems showing voltage dependence of channel opening approximate to that of the wild-type channel (*Jung et al., 2019*; *Kim et al., 2011*). Jung and colleagues found that the S273A mutant reduces the average whole-cell current densities to less than 50% of the wild-type (*Jung et al., 2019*), which indicates that S273A may be a risk factor for development of hearing loss through its limited ability to generate K$^+$ currents. However, whether F182L acts as a risk factor in DFNA2 hearing loss or should rather be considered a benign missense mutation (*Kim et al., 2011*) remains to be determined. We find that both the F182L and S273A mutants display an impaired response to DHA, which, if anything, is expected to preserve channel function in the presence of PUFAs.

To conclude, we find that hK$_V$7.4 and hK$_V$7.5 respond in opposing manners to PUFAs. The hK$_V$7.5 channel's response is largely in line with the responses that have previously been observed in other hK$_V$7 channels, whereas the hK$_V$7.4 channel response is not. Altogether, our study expands our understanding of how the activity of different members of the hK$_V$7 family are modulated by PUFAs and demonstrates different responses explained by different PUFA sites in different hK$_V$7 subtypes. Further

studies are needed to evaluate putative physiological importance of the effects described in this study in more complex experimental systems.

# Materials and methods

**Key resources table**

| Reagent type (species) or resource | Designation | Source or reference | Identifiers | Additional information |
|---|---|---|---|---|
| Gene (human) | KCNQ4 | GenBank | Acc.No. NM_004700 | |
| Gene (human) | KCNQ5 | GenBank | Acc.No. NM_001160133 | |
| Gene (human) | KCNE4 | GenBank | Acc.No. NM_080671 | |
| Chemical compound, drug | Tetradecatrienoic acid (TTA) | Larodan | Cat#: 10–1403 | |
| Chemical compound, drug | Hexadecatrienoic acid (HTA) | Larodan | Cat#: 10–1603 | |
| Chemical compound, drug | Linoleic acid (LA) | Sigma-Aldrich | Cat#: L1012 | |
| Chemical compound, drug | Arachidonic acid (AA) | Sigma-Aldrich | Cat#: A3611 | |
| Chemical compound, drug | Eicosapentaenoic acid (EPA) | Sigma-Aldrich | Cat#: E2011 | |
| Chemical compound, drug | Docosahexaenoic acid (DHA) | Sigma-Aldrich | Cat#: D2534 | |
| Chemical compound, drug | Docosahexaenoic acid methyl ester (DHA-me) | Sigma-Aldrich | Cat#: D2659 | |
| Chemical compound, drug | Indomethacin | Sigma-Aldrich | Cat#: I7378 | |

## Test compounds

All chemicals were purchased from Sigma-Aldrich, Stockholm, Sweden, unless stated otherwise. The PUFAs used in this study include: tetradecatrienoic acid (TTA, 14:3Δ5,8,11, Larodan, Stockholm, Sweden), hexadecatrienoic acid (HTA, 16:3Δ7,10,13, Larodan, Stockholm, Sweden), linoleic acid (LA, 22:6Δ4,7,10,13,16,19), arachidonic acid (AA, 20:4Δ5,8,11,14), eicosapentaenoic acid (EPA, 20:5Δ5,8,11,14,17), and docosahexaenoic acid (DHA, 22:6Δ4,7,10,13,16,19). PUFA analogues used in this study include: docosahexaenoic acid methyl ester (DHA-me), arachidonoyl amine (AA+), and docosahexaenoyl amine (DHA+). AA+ and DHA+ were synthesized *in house* as previously described (*Börjesson et al., 2010*). *Table 1* summarizes the molecular properties of PUFAs and PUFA analogues used. PUFAs and PUFA analogues were solved in 99.9% EtOH and diluted to their final concentrations in extracellular recording solution. To prevent potential degradation of arachidonic acid, recording solution was supplemented with 5 mM of the cyclooxygenase inhibitor indomethacin. Previous experiments with radiolabeled fatty acids have found that up to 30% of the nominal concentration applied will be bound to the Perspex recording chamber we use (*Börjesson et al., 2008*). As a result, the effective concentration of freely available fatty acids is 70% of the applied concentration. To allow for comparison with previous studies, we report the effective concentrations throughout the paper.

## Molecular biology

Human KCNQ4 (GenBank accession no. NM_004700), human KCNQ5 (GenBank accession no. NM_001160133) and human KCNE4 (GenBank accession no. NM_080671) were used in this study. cRNA for injection was prepared from DNA using a T7 mMessage mMachine transcription kit (Invitrogen, Stockholm, Sweden), and cRNA concentrations were determined by means of spectrophotometry (NanoDrop 2000c, Thermo Scientific, Stockholm, Sweden). Mutations were introduced through site-directed mutagenesis (QuikChange II XL, with 10 XL Gold cells; Agilent Technologies, Kista, Sweden) and confirmed by sequencing at the Linköping University Core Facility.

## Two-electrode voltage clamp experiments on *Xenopus* oocytes

Individual oocytes from *Xenopus laevis* frogs were acquired either through surgical removal followed by enzymatic digestion at Linköping University, or purchased from Ecocyte Bioscience (Dortmund, Germany). The use of animals, including the performed surgery, was reviewed and approved by the regional board of ethics in Linköping, Sweden (Case no. 1941). Oocytes at developmental stages V-VI

were selected for experiments and injected with 50 nL of cRNA. Each oocyte received either 2.5 ng of $hK_V7.4$ RNA or 5 ng of $hK_V7.5$ RNA. Co-injected oocytes received a 1:1 mix of $hK_V7.4$ cRNA (2.5 ng) and $hK_V7.5$ cRNA (2.5 ng) for the expression of $hK_V7.4/7.5$, or a mix of $hK_V7.4$ cRNA (2.5 ng) and hKCNE4 cRNA (1.25 ng) for the expression of $hK_V7.4/KCNE4$. Injected oocytes were incubated at 8 °C or 16 °C for 2–4 days prior to electrophysiological experiments.

Two-electrode voltage clamp recordings were performed with a Dagan CA-1B amplifier system (Dagan, MN, USA). Whole-cell $K^+$ currents were sampled using Clampex (Molecular devices, San Jose, CA, USA) at 5 kHz and filtered at 500 Hz. For most experiments, the holding potential was set to –80 mV. If experimental conditions allowed for channel opening at –80 mV, the holding potential was set to –100 mV. Current/voltage relationships were recorded using voltage-step protocols prior to and after the application of test compounds. Activation pulses were generated in incremental depolarizing steps of 10 mV, from –100 mV to +60 mV for $hK_V7.4$ and $hK_V7.4/KCNE4$, from –90 mV to 0 mV for $hK_V7.5$, and from –120 mV to +60 mV for $hK_V7.4/hK_V7.5$. The duration of the activation pulse was 2 s for $hK_V7.4$, $hK_V7.4/KCNE4$ and $hK_V7.4/7.5$, and 3 s for $hK_V7.5$. The tail voltage was set to –30 mV for all protocols and lasted for 1 s. All experiments were carried out at room temperature (approx. 20 °C). The extracellular recording solution consisted of (in mM): 88 NaCl, 1 KCl, 0.4 $CaCl_2$, 0.8 $MgCl_2$, and 15 HEPES. pH was adjusted to 7.4 by addition of NaOH. When experiments were conducted at a lower or higher pH, the pH was adjusted the same day as experiments by the addition of HCl or NaOH. Note that the pH of the extracellular recording solution was identical in the control recording and DHA recording of each oocyte. Recording solution containing test compounds was applied extracellularly to the recording chamber during an application protocol (comprised of repeated depolarizing steps every 10 s to 0 mV for $hK_V7.4$, $hK_V7.4/KCNE4$ or $hK_V7.4/hK_V7.5$ co-injected cells, or to –30 mV for $hK_V7.5$) until steady-state effects were observed. Solutions containing PUFAs or PUFA analogues were applied directly and manually to the recording chamber via a syringe. A minimum volume of 2 mL was applied to guarantee the replacement of the preceding solution in the recording chamber. The recording chamber was thoroughly cleaned between cells with 99.5% ethanol.

## Electrophysiological analysis

GraphPad Prism 8 software (GraphPad Software Inc, Ca, USA) was used for data analysis. The voltage-dependence of $hK_V7$ channels was approximated by plotting the immediate tail currents (recorded upon stepping to the tail voltage) against the preceding test voltages. Data were fitted with a Boltzmann function, generating a G(V) (conductance versus voltage) curve:

$$G_{(V)} = G_{min} + \frac{(G_{max} - G_{min})}{\left[1 + exp\left(\frac{V_{50} - x}{s}\right)\right]}$$

where $G_{min}$ is the minimum conductance, $G_{max}$ is the maximum conductance, $V_{50}$ is the midpoint of the curve (i.e, the voltage determined by the fit required to reach half of $G_{max}$) and s is the slope of the curve. The difference between $V_{50}$ under control settings and under test settings for each oocyte (the $\Delta V_{50}$) was used to quantify shifts in the voltage-dependence of channel opening evoked by test compounds. The relative difference between $G_{max}$ under control settings and under test settings for each oocyte (the $\Delta G_{max}$) was used to quantify changes in the maximum conductance evoked by test compounds. Note that representative G(V) curves have been normalized in figures to allow for better visualization of $V_{50}$ shifts.

To determine the concentration dependence or the pH dependence of $\Delta V_{50}$, the following concentration-response function was used:

$$\Delta V_{50} = \frac{\Delta V_{50,max}}{\left[1 + \left(\frac{EC_{50}}{C}\right)^N\right]}$$

where $\Delta V_{50,max}$ is the maximum shift in $V_{50}$, C is the concentration of the test compound, $EC_{50}$ is the concentration of a given test compound or the concentration of $H^+$ required to reach 50% of the maximum effect, and N is the Hill coefficient (set to 1 or –1). For studying pH dependence, the values of C were determined with asymptotic 95% confidence intervals in GraphPad Prism 8 and subsequently log-transformed to acquire the apparent pKa values.

## Coarse-grained molecular dynamics simulations

The full-length cryo-EM structure of $hK_V7.4$ channel in an intermediate activation state (PDB ID - 7BYL) (*Li et al., 2021*) was prepared by building in missing residues using the MODLOOP webserver (*Fiser et al., 2008*). The channel was embedded in a heterogenous bilayer consisting of 480 1-palmi toyl-2-oleoyl-sn-glycero-3- phosphocholine (POPC) molecules and 120 docosahexaenoic acid (DHA) molecules (placed at random and distributed equally in both the leaflets) using the CHARMM-GUI MARTINI (Bilayer system) maker (*Qi et al., 2015*). In one system, DHA with a negatively charged head group was used while in the other, DHA with a neutral head group was considered. The systems were then solvated by adding a~45 Å layer of water to each side of the membrane. Lastly, systems were ionized to reach a 150 mM KCl concentration. Martini 2.2 force field combined with ElNeDyn (Elastic network in dynamics) was used. The systems were minimized and equilibrated following the default CHARMM-GUI protocol. During equilibration, pressure was maintained at 1 bar through Berendsen pressure coupling; temperature was maintained at 300 K through velocity rescaling thermostat (*Bussi et al., 2007*) with the protein, membrane and solvent coupled. During production simulation, pressure was maintained at 1 bar through Parinello-Rahman pressure coupling (*Parrinello and Rahman, 1981*). A time step of 20 fs was used. Finally, 19 µs production simulations were carried out for the system with neutral DHA, while eight 5 µs simulations were run for the system with negatively charged DHA. Simulations were performed using GROMACS version 2020.4 (*Abraham et al., 2015*). The analysis of the coarse-grained simulations was carried out using PyLipid (*Song et al., 2022*). Frames extracted every 20 ns were used for the analysis. A dual cut-off scheme (lower limit: –0.5 nm, upper limit: –0.7 nm) was used to determine the interactions between DHA and the protein. Interactions were averaged across all subunits of the protein. Binding regions were determined by identifying clusters of at least four residues that interact with the same DHA molecule at the same time. These residues were found by community analysis of residue interaction networks. The most representative bound pose for each binding site was extracted by a scoring of the bound poses based on a density based scoring function, where the probability density is calculated from the simulation trajectories.

## Illustrations

For the structural images shown in *Figure 4*, the cryo-EM resolved structure of the $hK_V7.4$ channel (PDB ID - 7BYL [*Li et al., 2021*]) was visualized using The Protein Imager (*Tomasello et al., 2020*). Visualizations of the MD simulation systems were done using Visual Molecular Dynamics (VMD) (*Humphrey et al., 1996*).

## Statistical analysis

Average values are expressed as mean ± SEM. When comparing two groups, a Student's *t* test was performed. One sample *t* test was used to compare an effect to a hypothetical effect of 0 (for $\Delta V_{50}$) and 1 for ($\Delta G_{max}$). When comparing multiple groups, a one-way ANOVA was performed, followed by Dunnett's multiple comparison test when comparing to a single reference group. A p-value <0.05 was considered statistically significant. All statistical analyses were carried out in GraphPad Prism 8.

# Acknowledgements

We thank Dr H Peter Larsson, University of Miami, and Dr Fredrik Elinder, Linköping University, for comments on the manuscript. We thank Louise Abrahamsson for her contribution to experiments during her time as visiting scholar. The clones for human $K_V7.4$ and $K_V7.5$ were kind gifts from Dr. Nicole Schmitt at the University of Copenhagen. The clone for human KCNE4 was a kind gift from Dr. Bo H Bentzen at the University of Copenhagen. This project has received funding from the Swedish Society for Medical Research, the Science for Life Laboratory and the Swedish Research Council (2017–02040, 2018–04905 and 2021–01885).

# Additional information

### Competing interests

Lucie Delemotte: Reviewing editor, *eLife*. The other authors declare that no competing interests exist.

## Funding

| Funder | Grant reference number | Author |
|---|---|---|
| Swedish Research Council | 2017-02040 | Sara I Liin |
| Swedish Research Council | 2018-04905 | Lucie Delemotte |
| Swedish Research Council | 2021-01885 | Sara I Liin |
| Swedish Society for Medical Research | | Sara I Liin |

The funders had no role in study design, data collection and interpretation, or the decision to submit the work for publication.

## Author contributions

Damon JA Frampton, Koushik Choudhury, Conceptualization, Data curation, Formal analysis, Writing – original draft, Writing – review and editing; Johan Nikesjö, Data curation, Formal analysis, Writing – original draft, Writing – review and editing; Lucie Delemotte, Conceptualization, Data curation, Formal analysis, Funding acquisition, Supervision, Writing – original draft, Writing – review and editing; Sara I Liin, Conceptualization, Data curation, Formal analysis, Funding acquisition, Project administration, Supervision, Writing – original draft, Writing – review and editing

## Author ORCIDs

Damon JA Frampton 
Koushik Choudhury 
Johan Nikesjö 
Lucie Delemotte 
Sara I Liin 

## Ethics

The use of Xenopus laevis, including the performed surgery, was reviewed and approved by the regional board of ethics in Linköping, Sweden (Case no. 1941).

## Decision letter and Author response

Decision letter https://doi.org/10.7554/eLife.77672.sa1
Author response https://doi.org/10.7554/eLife.77672.sa2

# Additional files

## Supplementary files

• Transparent reporting form

## Data availability

Numerical data is provided in figures and/or corresponding figure legends, Table 2, and the source data files. Source data for molecular dynamics simulations is accessible at https://osf.io/6fuqs/.

The following dataset was generated:

| Author(s) | Year | Dataset title | Dataset URL | Database and Identifier |
|---|---|---|---|---|
| Delemotte L, Choudhury K | 2022 | Kv7 PUFAs | https://osf.io/6fuqs/ | Open Science Framework, 6fuqs |

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
