## [Editor Report]

In this manuscript, the authors describe the effects of polyunsaturated fatty acids in voltage-gated potassium channels of the Kv7 family that are specific for each subtype. The authors uncover a mechanism for this specificity which suggests that subtle structural differences can account for large effects, contributing to the physiological functions of these ion channels.

---

## [Decision Letter]

**Decision letter after peer review:**

[Editors’ note: the authors submitted for reconsideration following the decision after peer review. What follows is the decision letter after the first round of review.]

Thank you for submitting the paper "Subtype specific responses in hKv7.4 and hKv7.5 channels to polyunsaturated fatty acids" for consideration at *eLife*. Your submission has been reviewed by three peer reviewers, one of whom is a member of our Board of Reviewing Editors, and the evaluation has been overseen by a Senior Editor. Although the work is of interest, we are not convinced that the findings presented have the potential significance that we require for publication in *eLife*.

Specifically, upon reviewer consultation, the agreement was that the experimental design and execution are of very high quality and the findings are of interest, specially the seemingly non-electrostatic mechanism of modulation of the Kv7.4 channels by PUFAs, the manuscript lacks mechanistic insight into this new and potentially significant mode of modulation. The reviewers agree that if a set of experiments can be provided that probe the, presumably allosteric, mechanism of PUFA modulation of Kv7.4, a resubmission of a (new) manuscript to *eLife* will be encouraged.

*Reviewer #1:*

Modulation of some voltage-dependent potassium channels by unsaturated fatty acids seems to be mediated via an electrostatic mechanism by which the negative lipid head interacts with positive amino acid residues in the voltage-sensing domain or other amino acids responsible for producing a surface potential.

In this manuscript, Frampton et al. investigate the effect of omega-3 and 6 fatty acids on two Kv7 sub types, 7.4 and 7.5 for which data on modulation by these lipids was lacking.

In this manuscript, the authors present a series of interesting observations.

The main finding is that while Kv7.5 seems to be modulated by the established electrostatic mechanism, Kv7.4 is not modulated by a mechanism involving electrostatics. Rather, the authors find two unique phenylalanine residues in the S3 to be involved in the channel response to the fatty acid DHA, perhaps hinting at an allosteric effect.

While original, the findings are not further explored in terms of a mechanism, specifically the effects of fatty acids on Kv7.4.

For the most part, the experiments are well designed and executed. While the effects of DHA on Kv7.4 are robust, they are small and it remains to be seen if they have any physiological effect.

The new findings offer the possibility of studying new mechanisms of modulation of Kv7 channels mediated by unsaturated fatty acids.

1. Throughout the text, the authors refer to the effects of fatty acids (FAs) as activating or inhibiting. Have the authors shown that at negative voltages the channels (Kv7.5) can be opened by FAs? This result will justify calling FAs activators, otherwise, the effect should be called a modulation of activity. This is an important mechanistic distinction that in the absence of clear experimental evidence can be confusion.

2. The authors refer to a "local pH" on the channels that might account for the protonation state of DHA and thus its effect. It is not clear to me how the authors propose that the channel, which dos not conduct protons, can regulate local pH. Perhaps they want to say pKa?

3. While other experiments are well designed, the pH experiments intended to study the effect of charge on the modulation by DHA have a major flaw. There seem to be no controls for the effect of pH on channel gating (G-V curves). If this is the case, the measured effect of DHA is a combination of the possible effect of pH on DHA charge and the effect of pH on the GV of the channel.

4. The authors argue that the effects of FAs on Kv7 channels are mediated through the interaction of the negative charge of the head of FAs with gating charges in the VSD. But then, without any justification, carry out mutations of two (non-charged) phenylalanines present in the S3 of Kv7.4. The reference to gating charges should thus be removed.

*Reviewer #2:*

This manuscript by Frampton and colleagues provides a clear characterization of PUFA effects on the voltage dependence of Kv7.4 and Kv7.5 activation. The study finds interesting differences in PUFA's effects on these two Kv7 channels. PUFA shifts GV of Kv7.5 to left but of Kv7.4 to right; charges in the PUFA head are important to modulation of Kv7.5 but not so important to modulation of Kv7.4. The effect of PUFA on Kv7.5 G-V relation shows larger magnitudes than on Kv1-3 G-V relation at all pH values inspected. Mutations in Kv7.4 VSD, particularly F179 and F182 mutations in S3, alter PUFA effects. The data are of high quality and the description is clear. This study now adds to the literature on PUFA modulation of all Kv7 voltage dependence of activation. However, the manuscript provides a limited mechanistic insight on molecular interactions between the channels and the PUFA molecules. The study does not provide insights on how the modulation on Kv7 channels explains PUFA effects on physiology at the tissue level.

*Reviewer #3:*

In the present manuscript Damon J A Frampton et al. explore the effect of polyunsaturated fatty acids (PUFAs) on the hKv7.4 and hKv7.5 ion channels in a clear and well-presented manuscript. These subtype Kv7 channels, are found forming heterotetramers in vascular smooth muscle cells (Kv7.4/Kv7.5) or as homotetramers in cochlear outer hair cells (Kv7.4) their modulation by ligands or auxiliary KCNE subunits is of general physiological interest. Even though the effects of PUFAs have been extensively studied in other members of the Kv7 family (Kv7.1-3) hKv7.4 and hKv7.5 haven't been tested yet, therefore; the present manuscript fulfills that gap. These two Kv7 family subtypes are expressed as heterotetramers Kv7.4/7.5 in vascular smooth muscle cells, presumably as homotetramers (hKv7.4) in auditory cells Kv7.4. The results shown for hKv7.5 show a shift on the V50 of the G/V towards more negative potentials, and although the maximum conductance in presence of PUFAs seems to be unaffected, these results are compatible with a lipoelectric mechanism of interaction described previously for other members of the Kv7 family. The most novel and surprising finding of the present work is the inhibitory effect of PUFAs on hKv7.4, shifting the V50 of the G/V curves to more positive potentials, the opposite of what it has been observed in other members of the Kv7 family (including hKv7.5). Therefore, the mechanism for this effect is presumably different to the one described previously for other members of the family.

However, the authors have some hints about the effects of PUFAs on hKv7.4 in their data. The results modifying the charge of the DHA head group, suggest that a negative charge in the carboxy head group is important for the shift in the V50, and site directed mutagenesis highlight residues that modify discretely the PUFA effects on the V50shift. The double Phe mutations suggest a possible important site for PUFAs coordination by abolishing the PUFA V50 shift, reducing the current conductance, and slowing down the activation/deactivation kinetics (by visual inspection) moreover the double Phe mutant contains a disease associated mutation: F182L that on its own discretely affects the DHA effect on the V50 shift (still towards more positive potentials). However, it remains an open question what the effect of will be the second Phe, F179T on its own.

In these lines of trying to understand the PUFAs' mechanism of interaction in Kv7.4 there is some intriguing open questions for discussion, for example: is the coordination site for the PUFAs is still on the VSD involving the positive charges? are the two Phe in close proximity to any of these positive charges (located on S4 or S6) that have been shown to be important for Kv7.1 and conserved in 7.4 and 7.5?. Will the Tyr at the top of S5 mentioned by the authors in the Discussion (Yadzi et al. 2021) be also implicated in the PUFAs' effect?

I will encourage the authors to test F179T on its own to explore its effect on the PUFAs sensitivity.

Due to the intriguing effect of PUFAs and Kv7.4, and because KCNE subunits associate with Kv7 channels in multiple tissues, evaluating the effect of PUFAS in a Kv7.4-KCNE complex might be of interest. Specially since these auxiliary subunits affect the Kv7 channels response to PUFAs, as shown by the authors (Liin SI, Silverå Ejneby M, Barro-Soria R, et al. Polyunsaturated fatty acid analogs act antiarrhythmically on the cardiac IKs channel. Proc Natl Acad Sci U S A. 2015;112(18):5714-5719. doi:10.1073/pnas.1503488112) where the KCNE1 presence in the Kv7.1/KCNE1 complex abolishes the V50 shift produced by DHA with Kv7.1 alone. I will suggest the authors to consider trying to express Kv7.4 together with a member of the KCNE family, to test the effects on PUFAs sensitivity. (Jepps, Thomas A et al. "Fundamental role for the KCNE4 ancillary subunit in Kv7.4 regulation of arterial tone." The Journal of physiology vol. 593,24 (2015): 5325-40. doi:10.1113/JP271286).

As a general comment for all the figures, I would suggest the authors to show averages instead of representative experiments for the G/Vs. Some of the examples shown are not representative to the averages in Table II. For example: Table II Kv7.4 WT control V_50_ = –11.1mV. Figure 1 (legend) for the same condition: (V_50_ = –4.8mV).

I assume that the change in the slope as well as the V50 shift in presence of DHA reported in lines 105-115 were obtain from normalized currents, however the normalization of the currents is mentioned at the end of this paragraph. Please adjust the text.

Since one of the two Phe mutated is F182L which is also a disease associated mutation I will suggest moving the data on the disease associated mutations to the previous section with the double Phe mutant for clarity.

What is the "n" on the Supplementary Figure?

Figure 4 panel F: the voltage protocol is missing.

Please cite the structure of the Kv7.4 when introduce the phenylalanine mutants (line 235).

[Editors’ note: further revisions were suggested prior to acceptance, as described below.]

Thank you for resubmitting your work entitled "Subtype-specific responses of hKv7.4 and hKv7.5 channels to polyunsaturated fatty acids reveal an unconventional modulatory site and mechanism" for further consideration by *eLife*. Your revised article has been evaluated by Richard Aldrich (Senior Editor) and a Reviewing Editor.

The manuscript has been improved but there are some remaining issues that need to be addressed, as outlined below:

After consultation, the reviewers agree that the manuscript is greatly improved and that the authors have made a great effort to further study the mechanism of the unique action pf PUFAs on kv7.4 channels. While the manuscript now provides experiments and simulations that suggest a unique mechanism of interaction of PUFA with inner-facing S4 positive charges, there are some possible interpretations of the results that need to be discussed further.

The authors observe in their experiments an effect with negative charged PUFAs and no effect with the non-charged/positive ones. The gating charges mutated at the bottom of S4 are non-specific of Kv7.4 so this effect seen in the MD simulations could be "charge-driven" instead of Kv7.4 specific. The authors should add to the discussion of why they think this effect is specific on Kv7.4 comparing the Kv7.1 and Kv7.4 structures. While these questions could be addressed with MD simulations with the Kv7.1, these are not necessary to answer the current concerns.

Please also address the specific concern regarding the quantitation of the pH effects of PUFA modulation of these channels.

*Reviewer #1:*

This is a resubmission of a previous manuscript. In this new version, the authors have made a great effort to address that comments raised in the first round of reviews. For the most part, the paper is greatly improved and it now presents new experiments and simulations that provide a mechanistic explanation for the diverging effects of PUFAs observed in the two types of Kv7.5 and Kv7.4 channels.

While most of the previous concerns have been dealt with, I still have an objection to the presentation and interpretation of the pH effects on the DHA modulation of the vhalf of kv7.1 and kv7.5 channels. The change in extracellular pH will effectively change the protonation state of DHA and also have an effect on the voltage-dependent gating of the channel (via surface charge or specific proton effects). Thus the reported shifts, as they are analyzed in figure 5 also include the shift due to gating. The authors should do control experiments of the pH effect on the Vhalf, in the absence of DHA and calculate the effect of protonation of DHA as the δ δ Vhalf.

*Reviewer #2:*

First of all, I would like to thank the authors for their effort in addressing my comments and concerns in this exciting new version of the manuscript. I would also like to acknowledge their effort in providing a novel mechanism that can explain the fascinating subtype-specific effect of the PUFAS on Kv7.4. The newly added molecular dynamic (MD) simulations beautifully predict a putative PUFA binding site in the inner part of S4 that the authors further confirmed with a set of functional experiments. The results of the site-directed mutagenesis of the positive charge of the lower VSD suggests a conserved lipoelectric conventional PUFA binding site in Kv7.4 that is overruled by the unconventional "inner VSD site" one. This unique mechanism opens the door to new and exciting questions about its relevance and physiological role.

The latest additions to the Results section explain the uniqueness of the PUFAs effect on Kv7.4 and provide a mechanism. I have just a couple of suggestions to reinforce the specificity of the results. First, I wonder if the new arginine mutants are only modulated by negatively charged DHA or non-charged/positive charged PUFAS also modify the channel, as one could expect if they are acting through the conventional lipoelectric PUFA binding site. Additionally, to help support the particularity of the new "inner VSD site" Kv7.4, the authors could take advantage of the Kv7.1 structure and perform the same type of MD simulations to confirm further that the "inner VSD site" for negatively charged DHA found in Kv7.4 is specific as it is the shift of the G-V curves towards positive potentials.

---

## [Author Response]

[Editors’ note: the authors resubmitted a revised version of the paper for consideration. What follows is the authors’ response to the first round of review.]

Reviewer #1:[…]1. Throughout the text, the authors refer to the effects of fatty acids (FAs) as activating or inhibiting. Have the authors shown that at negative voltages the channels (Kv7.5) can be opened by FAs? This result will justify calling FAs activators, otherwise, the effect should be called a modulation of activity. This is an important mechanistic distinction that in the absence of clear experimental evidence can be confusion.

We have now adjusted the text to make clear that PUFAs modulate channel activity. We now refer to the effects as facilitation or inhibition of channel activity.

2. The authors refer to a "local pH" on the channels that might account for the protonation state of DHA and thus its effect. It is not clear to me how the authors propose that the channel, which dos not conduct protons, can regulate local pH. Perhaps they want to say pKa?

We have previously shown for Kv7.1 that the local pH environment in a PUFA binding site is altered upon Kv7.1 co-assembly with the KCNE1 subunit (Larsson et al., *eLife* 2018). That alteration is not caused by protons conducted through the channel pore but rather through structural rearrangements that places one of the negatively charged extracellular loops of Kv7.1 closer to the PUFA head group. This structural re-arrangement is proposed to attract positively charged protons to the PUFA binding site, thereby promoting DHA protonation (and increasing the apparent pKa of the DHA head group). The functional outcome of KCNE1 co-assembly with Kv7.1 is a reduced DHA response. In a similar manner, we hypothesized that Kv7.5 might show an increased DHA response by instead having a local pH environment in the PUFA binding site that promotes DHA deprotonation. We envision that this could be caused by the presence of, for instance more positively charged residues near the DHA headgroup compared to in Kv7.1 and Kv7.2/7.3. However, as shown by the experimental data (Figure 5C), this was not the case as the apparent pKa of DHA was similar for both Kv7.1 and Kv7.5. We have adjusted the text in the Results section to rather refer to altered apparent pKa, to avoid confusion.

3. While other experiments are well designed, the pH experiments intended to study the effect of charge on the modulation by DHA have a major flaw. There seem to be no controls for the effect of pH on channel gating (G-V curves). If this is the case, the measured effect of DHA is a combination of the possible effect of pH on DHA charge and the effect of pH on the GV of the channel.

We apologize that the design of the pH experiments was not clearly described. In all experiments, the extracellular pH of the external solution is similar in the control recording and the DHA recording. This is because the pH of the extracellular recording solution was set the day of experiments, and DHA was solved at the desired concentration in this extracellular recording solution. For example, in the experiment shown in Figure 5B, the pH of the extracellular solution was set to pH 8.2, and was used in both the control measurement (open circles) and DHA measurement (red circles) as DHA was solved in the pH 8.2 recording solution. Therefore, the measured effect of DHA reports only on the shift induced by DHA (and not by pH alone). We have now clarified this in the methods section and appropriate figure legend.

4. The authors argue that the effects of FAs on Kv7 channels are mediated through the interaction of the negative charge of the head of FAs with gating charges in the VSD. But then, without any justification, carry out mutations of two (non-charged) phenylalanines present in the S3 of Kv7.4. The reference to gating charges should thus be removed.

We apologize that the justification was unclear. What we meant was that structural differences in the VSD of Kv7.4 compared to other Kv7 channels, induced for instance by bulky phenylalanines, may allosterically impair conventional PUFA interaction with the outermost gating arginines in the VSD. We have in the revised version of the manuscript removed this Results section and the data for the double phenylalanine mutant. Instead, we now show molecular dynamics and simulation data supporting a unique, unconventional, PUFA site in Kv7.4 underlying PUFA-induced inhibition of channel activation. Our new data suggest that PUFAs, like in Kv7.1 channels, have several putative interaction sites on Kv7.4. However, notably, we find a functionally dominating and unconventional site in Kv7.4 not previously observed in other Kv7 channels. This Kv7.4 PUFA site is in the inner half of the voltage-sensing domain and is characterized by interactions between the negatively charged PUFA head group and positively charged arginine residues in the lower half of S4 (the innermost gating charges). Experimental neutralization of these innermost arginines ablates the unconventional PUFA-induced inhibition of Kv7.4 and endows Kv7.4 with a conventional PUFA response (i.e. a shifted V50 towards more negative voltages). We propose that PUFA binding to the unconventional inner VSD site in Kv7.4 shifts V50 towards more positive voltages by stabilizing resting and/or intermediate S4 conformations (thus hindering channel opening), and that disruption of this site allows for PUFAs to instead facilitate Kv7.4 channel opening by acting from the conventional outer VSD site. Altogether, these new data presented in two new Results sections (rows 288-332), Figures 6-7 with Figure Supplements, and discussed in one new section (rows 389-406) provides a mechanistic basis for the unconventional PUFA response of Kv7.4.

Speculatively, the two phenylalanines assessed in the previous version of the manuscript contributes to the slightly altered packing of the VSD of Kv7.4 required to enable PUFA binding to the unconventional inner PUFA site. This will be explored in future work.

Reviewer #2:This manuscript by Frampton and colleagues provides a clear characterization of PUFA effects on the voltage dependence of Kv7.4 and Kv7.5 activation. The study finds interesting differences in PUFA's effects on these two Kv7 channels. PUFA shifts GV of Kv7.5 to left but of Kv7.4 to right; charges in the PUFA head are important to modulation of Kv7.5 but not so important to modulation of Kv7.4. The effect of PUFA on Kv7.5 G-V relation shows larger magnitudes than on Kv1-3 G-V relation at all pH values inspected. Mutations in Kv7.4 VSD, particularly F179 and F182 mutations in S3, alter PUFA effects. The data are of high quality and the description is clear. This study now adds to the literature on PUFA modulation of all Kv7 voltage dependence of activation. However, the manuscript provides a limited mechanistic insight on molecular interactions between the channels and the PUFA molecules. The study does not provide insights on how the modulation on Kv7 channels explains PUFA effects on physiology at the tissue level.

We have now performed molecular dynamics simulations paired with site-directed mutagenesis and electrophysiology experiments to probe the mechanism of PUFA modulation of Kv7.4. In brief, our new data suggest that PUFAs, like in Kv7.1 channels, have several putative interaction sites on Kv7.4. However, notably, we find a functionally dominating and unconventional site in Kv7.4 not previously observed in other Kv7 channels. This Kv7.4 PUFA site is in the inner half of the voltage-sensing domain and is characterized by interactions between the negatively charged PUFA head group and positively charged arginine residues in the lower half of S4 (the innermost gating charges). Experimental neutralization of these innermost arginines ablates the unconventional PUFA-induced inhibition of Kv7.4 and endows Kv7.4 with a conventional PUFA response (i.e. a shifted V50 towards more negative voltages). We propose that PUFA binding to the unconventional inner VSD site in Kv7.4 shifts V50 towards more positive voltages by stabilizing resting and/or intermediate S4 conformations (thus hindering channel opening), and that disruption of this site allows for PUFAs to instead facilitate Kv7.4 channel opening by acting from the conventional outer VSD site. Altogether, these new data presented in two new Results sections (rows 288-332), Figures 6-7 with Figure Supplements, and discussed in one new section (rows 389-406) provides a mechanistic basis for the unconventional PUFA response of Kv7.4.

Reviewer #3:[…]I will encourage the authors to test F179T on its own to explore its effect on the PUFAs sensitivity.

We thank the reviewer for the suggestion for further experiments. We have tested the proposed F179T mutation on hK_V_7.4 and found that, similar to the two mutations associated with impaired hearing, the F179T mutation impaired the inhibitory hK_V_7.4 response to DHA (ΔV_50_ by 70 μM of DHA = -1.3 ± 2.2 mV). However, we have in the revised version of the manuscript removed the Results section for the double phenylalanine mutant in favour of new data suggesting a novel Kv7.4 PUFA binding site, as described in the response to Reviewer 1. Speculatively, the two phenylalanines assessed in the previous version of the manuscript contributes to the slightly altered packing of the VSD of Kv7.4 required to enable PUFA binding to the unconventional inner PUFA site. This will be explored in future work.

Due to the intriguing effect of PUFAs and Kv7.4, and because KCNE subunits associate with Kv7 channels in multiple tissues, evaluating the effect of PUFAS in a Kv7.4-KCNE complex might be of interest. Specially since these auxiliary subunits affect the Kv7 channels response to PUFAs, as shown by the authors (Liin SI, Silverå Ejneby M, Barro-Soria R, et al. Polyunsaturated fatty acid analogs act antiarrhythmically on the cardiac IKs channel. Proc Natl Acad Sci U S A. 2015;112(18):5714-5719. doi:10.1073/pnas.1503488112) where the KCNE1 presence in the Kv7.1/KCNE1 complex abolishes the V50 shift produced by DHA with Kv7.1 alone. I will suggest the authors to consider trying to express Kv7.4 together with a member of the KCNE family, to test the effects on PUFAs sensitivity. (Jepps, Thomas A et al. "Fundamental role for the KCNE4 ancillary subunit in Kv7.4 regulation of arterial tone." The Journal of physiology vol. 593,24 (2015): 5325-40. doi:10.1113/JP271286).

We agree with the reviewer that assessing the PUFA response of Kv7.4 co-expressed with KCNE4 is of interest. We now include data for oocytes co-expressing Kv7.4 and KCNE4 in a new Results section (rows 228-236) and Figure 4B-C. These experiments show that the DHA effect on Kv7.4 is not affected by the KCNE4 subunit, which responded to DHA with a positive shift in V50 comparable to that of Kv7.4 alone. Thus, the KCNE4 subunit did not alter the Kv7.4 response to DHA, which is different to what we have previously found for the KCNE1 subunit and Kv7.1. However, given the different PUFA binding sites underlying Kv7.4 and Kv7.1 effects, respectively, and the clear contrast in how the two KCNE subunits alter channel function, this is not surprising.

Note that the currents generated by co-expressed Kv7.4 and KCNE4 were relatively small. The reason was that in our hands, oocytes co-expressing Kv7.4 and KCNE4 tended to result in fragile and leaky oocytes. Thus, reliable determination of Kv7.4/KCNE4 channel pharmacology could only be done in oocytes with relatively low channel expression.

As a general comment for all the figures, I would suggest the authors to show averages instead of representative experiments for the G/Vs. Some of the examples shown are not representative to the averages in Table II. For example: Table II Kv7.4 WT control V50 = –11.1mV. Figure 1 (legend) for the same condition: (V50 = –4.8mV).

We thank the reviewer for the suggestion of showing averages instead of representative G(V) curves. However, as the reviewer points out, there is cell-to-cell variability in the intrinsic V50 under control conditions, which is part of the natural variability in our biological data set. In our experience, averaging G(V) curves with such variability in intrinsic V50 makes it more difficult to visualize/appreciate shifts in V50 induced by DHA (because the DHA-induced shift in V50 is in part masked by the natural variability in V50). If possible, we would therefore prefer showing representative examples instead of averages. We have now included another representative example of Kv7.4 WT in Figure 1, with a V50 more similar to the average V50.

I assume that the change in the slope as well as the V50 shift in presence of DHA reported in lines 105-115 were obtain from normalized currents, however the normalization of the currents is mentioned at the end of this paragraph. Please adjust the text.

We apologize for not describing this in a clear manner. Normalization is done only in figures for illustrative reasons to make it easier to see shifts. In contrast, our data analysis is done on data that has not been normalized. We have now adjusted the text to clarify this.

Since one of the two Phe mutated is F182L which is also a disease associated mutation I will suggest moving the data on the disease associated mutations to the previous section with the double Phe mutant for clarity.

As described above, data for the double phenylalanine mutant has been removed from the revised version of the manuscript. The data for the disease associated mutations have now been moved to the new Figure 4.

What is the "n" on the Supplementary Figure?

Supplementary Figure 1 showed a representative recording each for Kv7.4 and Kv7.5 (i.e. n = 1). We have adjusted the figure legend to clarify this.

Figure 4 panel F: the voltage protocol is missing.

Thank you for pointing this out. We have now included voltage protocols in all panels showing current families.

Please cite the structure of the Kv7.4 when introduce the phenylalanine mutants (line 235).

We now cite the structure of Kv7.4 when first mentioned.

[Editors’ note: what follows is the authors’ response to the second round of review.]

Reviewer #1:This is a resubmission of a previous manuscript. In this new version, the authors have made a great effort to address that comments raised in the first round of reviews. For the most part, the paper is greatly improved and it now presents new experiments and simulations that provide a mechanistic explanation for the diverging effects of PUFAs observed in the two types of Kv7.5 and Kv7.4 channels.While most of the previous concerns have been dealt with, I still have an objection to the presentation and interpretation of the pH effects on the DHA modulation of the vhalf of kv7.1 and kv7.5 channels. The change in extracellular pH will effectively change the protonation state of DHA and also have an effect on the voltage-dependent gating of the channel (via surface charge or specific proton effects). Thus the reported shifts, as they are analyzed in figure 5 also include the shift due to gating. The authors should do control experiments of the pH effect on the Vhalf, in the absence of DHA and calculate the effect of protonation of DHA as the δ δ Vhalf.

We thank the reviewer for their kind words, and apologize for a lack of clarity in the description of our pH effects.

We have now included a supplementary figure (Figure 5 —figure supplement 1), to show V50 values without and with DHA at different pH and to clarify how the V50 was calculated. The black data points represent V50 of hKv7.5 in control solution with indicated pH, which shows that the control V50 values are the same despite the pH being titrated from acidic to alkaline pH values. Thus, for hKv7.5, titrating the pH of the control solution from 6.5 to 8.2 has no effect on V50. This is different from our experience of working with hKv7.1, for which V50 shifts by up to 20 mV when titrating the pH of the control solution from 6.5 to 8.2.

The red data points represent V50 of hKv7.5 in DHA-supplemented control solution with different pH, which shows that the ability of DHA to shift V50 towards more negative voltages varies with pH. DHA has no effect on V50 at pH 6.5 (the red and black data points overlap), whereas at pH 8.2 DHA shifts V50 more than 40 mV towards more negative voltages. The DHA-induced shift (∆V50) plotted in Figure 5C is the difference between V50 in the absence and presence of DHA (black and red data points, respectively) for each pH. We hope that the new Figure 5 —figure supplement 1, referred to at rows 272-279, together with its clarifying figure legend addresses any concerns that the reviewer may have in regard to the pH effects we observe.

Reviewer #2:First of all, I would like to thank the authors for their effort in addressing my comments and concerns in this exciting new version of the manuscript. I would also like to acknowledge their effort in providing a novel mechanism that can explain the fascinating subtype-specific effect of the PUFAS on Kv7.4. The newly added molecular dynamic (MD) simulations beautifully predict a putative PUFA binding site in the inner part of S4 that the authors further confirmed with a set of functional experiments. The results of the site-directed mutagenesis of the positive charge of the lower VSD suggests a conserved lipoelectric conventional PUFA binding site in Kv7.4 that is overruled by the unconventional "inner VSD site" one. This unique mechanism opens the door to new and exciting questions about its relevance and physiological role.The latest additions to the Results section explain the uniqueness of the PUFAs effect on Kv7.4 and provide a mechanism. I have just a couple of suggestions to reinforce the specificity of the results. First, I wonder if the new arginine mutants are only modulated by negatively charged DHA or non-charged/positive charged PUFAS also modify the channel, as one could expect if they are acting through the conventional lipoelectric PUFA binding site. Additionally, to help support the particularity of the new "inner VSD site" Kv7.4, the authors could take advantage of the Kv7.1 structure and perform the same type of MD simulations to confirm further that the "inner VSD site" for negatively charged DHA found in Kv7.4 is specific as it is the shift of the G-V curves towards positive potentials.

We thank the reviewer for their kind words and their suggestions of how to reinforce our conclusions.

We have now performed experiments on the arginine mutants (R213Q, R216Q and R219Q) with the uncharged PUFA analogue DHA-me, and with two positively charged PUFA analogues (DHA+ and AA+) (Figure 7 —figure supplement 1). Our results indicate that the PUFA response of the arginine mutants can be electrostatically tuned. The response pattern was clearest for R213Q, which showed the largest negative shift induced by DHA and for which DHA-me had minimal effects on V_50_ whereas DHA+ and AA+ shifted V_50_ towards positive voltages. The response pattern showed a similar trend for R216Q and R219Q, however, with less robust effects induced by the positively charged compounds. The positive shifts induced by AA+ on the arginine mutants were larger than for hK_V_7.5 (Figure 3C) and those previously reported for the Shaker K_V_ channel (Börjesson et al., 2010), comparable to the effect on WT hK_V_7.4 (Figure 3F), and slightly smaller than those previously reported for hK_V_7.1 (Liin et al., 2015). These data, provided in Figure 7 – Figure Supplement 1 and described at rows 334-349 and 423-427, further indicate that the PUFA responses of the Kv7.4 arginine mutants are more in line with the lipoelectric mechanism that has been proposed for other Kv7 channels.

In regards to why the unconventional inner VSD site may be specific to hKv7.4, we have now included a structural alignment of the VSDs of hKv7.4 and hKv7.1 (Figure 6 —figure supplement 3). This alignment shows two possible molecular level reasons as to why PUFAs may be permitted to interact with this site in hKv7.4, which are now described in the discussion of the paper (rows 427-433) as follows:

“A structural alignment between the VSDs of hK_V_7.1 (PDB ID – 6UZZ) and hK_V_7.4 (PDB ID – 7BYL) revealed two possible molecular level reasons for the differential binding of DHA to these two channels: in hK_V_7.4 the binding site we uncovered indeed appears both larger, with the S4 helix pushed inwards in hK_V_7.1 relative to hK_V_7.4, and more accessible, as the cleft between S1 and S2 via which DHA appears to enter the cavity features a bulky, obstructing phenylalanine residue (F166) in hK_V_7.1 compared to a smaller valine residue (V142) in hK_V_7.4 (Figure 6 —figure supplement 3).”